# Effects of exogenous β-glucanase on ileal digesta soluble β-glucan molecular weight, digestive tract characteristics, and performance of coccidiosis vaccinated broiler chickens fed hulless barley-based diets with and without medication

**Namalika D. Karunaratne[1], Rex W. Newkirk[1]\*, Nancy P. Ames[2], Andrew G. Van Kessel[1], Michael R. Bedford[3], Henry L. Classen[1]**

1 Department of Animal and Poultry Science, University of Saskatchewan, Saskatoon, Saskatchewan, Canada, 2 Agriculture and Agri-Food Canada, Winnipeg, Manitoba, Canada, 3 AB Vista, Marlborough, Wiltshire, United Kingdom

\* rex.newkirk@usask.ca

## Abstract

### Introduction

Limited use of medication in poultry feed led to the investigation of exogenous enzymes as antibiotic alternatives for controlling enteric disease. The objective of this study was to evaluate the effects of diet β-glucanase (BGase) and medication on β-glucan depolymerization, digestive tract characteristics, and growth performance of broilers.

### Materials and methods

Broilers were fed hulless barley (HB) based diets with BGase (Econase GT 200P from AB Vista; 0 and 0.1%) and medication (Bacitracin and Salinomycin Na; with and without) arranged as a 2 × 2 factorial. In Experiment 1, 160 broilers were housed in cages from d 0 to 28. Each treatment was assigned to 10 cages. In Experiment 2, broilers (2376) were housed in floor pens and vaccinated for coccidiosis on d 5. Each treatment was assigned to one floor pen in each of nine rooms.

### Results

In Experiment 1, the soluble β-glucan weighted average molecular weight (Mw) in the ileal digesta was lower with medication in the 0% BGase treatments. Peak molecular weight (Mp) and Mw were lower with BGase regardless of medication. The maximum molecular weight for the smallest 10% β-glucan (MW-10%) was lower with BGase addition. In Experiment 2, Mp was lower with medication in 0% BGase treatments. Beta-glucanase resulted in lower Mp regardless of medication, and the degree of response was lower with medication. The MW-10% was lower with BGase despite antibiotic addition. Body weight gain and feed

**Data Availability Statement:** All relevant data are within the paper and its Supporting Information files.

**Funding:** This study was funded by the following grants to HLC: Aviagen North America (# 414702), Sofina Foods Inc. (# 414705), Prairie Pride Natural Foods Ltd. (# 414707), Chicken Farmers of Saskatchewan (# 414704), Canadian Poultry Research Council (#414703), Poultry Industry Council, Canada (# 414706), Saskatchewan Broiler Hatching Egg Producer's Marketing Board, Saskatchewan Egg Producers and Saskatchewan Turkey Producers' Marketing Board (# 414708), Saskatchewan Egg Producers (# 414709), Saskatchewan Turkey Producers' Marketing Board (# 414710), University of Saskatchewan (# 414739), and National Science and Engineering Research Council (NSERC) (#414764). AB Vista provided support in the form of salaries for author Michael R. Bedford. The specific roles of these authors are articulated in the 'author contributions' section. The funders had no role in the study design, data collection and analysis, decision to publish, or preparation of the manuscript.

**Competing interests:** The authors have read the journal's policy and the authors of this manuscript have the following competing interests: MRB is a paid employee of AB Vista. This study was funded by Aviagen North America, Sofina Foods Inc., Prairie Pride Natural Foods Ltd., Chicken Farmers of Saskatchewan, Canadian Poultry Research Council, Poultry Industry Council, Canada, Saskatchewan Broiler Hatching Egg Producer's Marketing Board, Saskatchewan Egg Producers, Saskatchewan Turkey Producers' Marketing Board, University of Saskatchewan, and National Science and Engineering Research Council (NSERC). This does not alter our adherence to PLOS ONE policies on sharing data and materials. There are no patents, products in development or marketed products to declare.

efficiency were higher with medication regardless of BGase use through-out the trial (except d 11–22 feed efficiency). Beta-glucanase resulted in higher body weight gain after d 11 and worsened and improved feed efficiency before and after d 11, respectively, in unmedicated treatments.

## Conclusion

BGase and medication caused the depolymerization of soluble ileal β-glucan. Beta-glucanase acted as a partial replacement for diet medication by increasing growth performance in coccidiosis vaccinated broilers.

## Introduction

Antibiotics have been used in poultry feed at sub-therapeutic doses for decades to improve growth and feed efficiency and prevent enteric infections [1]. However, the prolonged and indiscriminate use of antimicrobials in animal production is likely to cause antibiotic resistance in pathogenic bacteria. Its effect on animal and human health risk has led to reduced use of in-feed antibiotics in the poultry industry [2, 3]. Potential alternatives to antibiotics that have been studied include probiotics, prebiotics, organic acids, essential oils, and feed enzymes [4, 5].

Prebiotics are non-digestible feed ingredients that beneficially affect the host by selectively stimulating the growth and function of beneficial microbiota in the digestive tract [6]. The most commonly available prebiotics are oligosaccharides from various sources and small molecular weight polysaccharides derived from cereal grains. Dietary inclusion of arabinoxylo-oligosaccharides/ xylo-oligosaccharides affects gastro-intestinal microbial populations of chickens by increasing beneficial bacteria, including Bifidobacteria, Lactobacilli and *Clostridium* cluster XIV [7, 8], and reducing *Salmonella* colonization in the caeca and translocation to the spleen [9]. In addition, exogenous xylanase in wheat-based diets increased the number of gastro-intestinal beneficial bacteria, including lactic acid bacteria, while reducing pathogenic bacteria in broiler chickens [10, 11], possibly by decreasing the molecular weight of soluble arabinoxylan derived from the wheat. Arabinoxylan has been extensively studied concerning its ability to act as a prebiotic since arabinoxylan is found in the cell walls of the most common cereals used in poultry feed (wheat and corn), and prebiotic oligosaccharides are presumed to be formed by the use of dietary xylanase. However, research is limited regarding cereal β-glucan since it predominates in barley and oats, which are less commonly found in poultry feed.

Hulless barley (HB) contains a higher level of β-glucan than conventional barley due to the removal of the hull during processing [12, 13]. Further, many HB cultivars are developed for the human food industry, and as a result, are selected for high β-glucan content [14]. Dietary enzymes such as endo-β-glucanase depolymerize larger molecular weight β-glucan producing lower molecular weight compounds, which are fermentable in the distal digestive tract [15]. A consequence of fermentation is the production of short-chain fatty acids (SCFA), which are thought to improve digestive tract morphology and physiology and stimulate the establishment of beneficial bacterial populations while at the same time reducing colonization by pathogens [15, 16]. However, the effects of exogenous BGase on microbial fermentation and digestive tract physiology and morphology are less-well studied, and the results have been inconsistent in previous research.

Feed medication mechanisms are not fully understood, although antibiotics have been successfully used to promote growth and feed efficiency and improve bird health [17, 18]. The

primary mechanism is generally accepted as a positive modulation of the diversity and relative abundance of bacteria in the digestive tract microbial community, and thereby the control of enteric disease and stimulation of immune function in broiler chickens [19–21]. However, another mechanism of action is the direct anti-inflammatory activity of antibiotics [22]. Investigating the interaction between medication and enzyme use in high fibre diets offers the potential to add knowledge on medication mechanisms of action and study the effectiveness of enzymes in reducing the adverse effects of enteric disease. The effects of exogenous BGase and diet medication on broiler performance and digestive tract characteristics could depend on the age of the birds due to the distinct maturity of the digestive tract, including the development of gut microbiota, and housing conditions that affect the level of exposure to pathogenic organisms. Therefore, the current study utilized the same experimental design and treatments in two different environments.

The objective of the current study was to investigate the effects of exogenous BGase and medication on ileal digesta soluble β-glucan molecular weight, digestive tract characteristics, and production performance of broiler chickens fed an HB-based diet under different housing environments and disease conditions. Experiment 1 was completed in cages, and the birds had a less exposure to pathogenic microbes and lower ability of coprophagy due to the clean environment. Experiment 2 was completed in litter floor pens using broilers vaccinated for coccidiosis and raised at high humidity and litter moisture that increase coccidia cycling. The rationale for these experiments was to determine if treatments produce the same effects in the two experiments that contained different housing environments and microbial exposure. It was hypothesized that exogenous BGase would depolymerize high molecular weight β-glucan, resulting in increased fermentation and beneficial effects on digestive tract morphology and physiology. This should result in improved performance of broiler chickens and reduce the requirement for the medication in broilers fed HB-diets. Further, a higher response to exogenous BGase and a greater reduction of diet medication necessity would be expected from the broiler chickens from Experiment 2 (coccidiosis-vaccinated) compared to Experiment 1.

## Materials and methods

The experimental procedure was approved by the Animal Research Ethics Board of the University of Saskatchewan and conducted according to the Canadian Council on Animal Care guidelines for humane animal use [23, 24].

### Experiment 1

**Birds and housing.**    A total of 160 broiler chickens (Ross × Ross 308) were obtained from a commercial hatchery on the day of hatch and housed in battery cages (length, 51 cm; width, 51 cm; height, 46 cm). The chickens were kept in thermal comfort, and the day length was reduced from 23 h at d 0 to 18 h at d 8. Birds were given feed and water *ad-libitum*. There were 10 cage replications per treatment and four birds per cage. Treatments were randomly assigned to the battery cages.

**Experimental diets.**    The dietary treatments were arranged according to a 2 × 2 factorial arrangement (BGase and medication). Beta-glucanase (Econase GT 200 P from ABVista, Wiltshire, UK) levels were 0 and 0.1% (the BGase activity of 0 and 200,000 BU/kg, respectively), and diets were fed without or with medication; Bacitracin (Zoetis Canada Inc., Kirkland, QC, Canada) at 4.4 mg/kg and Salinomycin Sodium (Phibro Animal Health Corporation, Teaneck, NJ) at 25 mg/kg. Diets were based on 60% HB (CDC Fibar) and were formulated to meet or exceed Ross 308 broiler nutrition specifications [25]. The ingredients and calculated nutrient levels are shown in Table 1, and the diets were fed in crumble form. The pelleting temperature

was controlled between 70–75˚C to prevent high temperature-induced BGase inactivation during feed processing. Measured BGase activity in diets approached the estimated values, thereby confirming BGase was added correctly and that activity was not lost during feed processing. Xylanase activity was non-detectable in experimental diets.

**Rearing performance data collection.** Body weight and feed intake were measured on a cage basis at d 7, 14, 21 and 28. The birds were checked two times (morning and evening) daily for health and behavior throughout the study. The specific criteria used to determine humane endpoints included yolk sac infection with apparent distress, starve-out, runt and other issues with bird mobility that compromises the ability to eat and drink, weight gain, and the obvious changes with no chance of recovery. Mortality was recorded daily, and dead birds were sent to Prairie Diagnostic Services for necropsy.

**Sample collection.** All birds were euthanized on d 28 by administering T-61 (Merck animal health, Kirkland, Quebec, Canada) into the brachial vein. Birds were weighed individually. Two birds per cage were used for pH measurement and to collect samples for SCFA analysis. *In-situ* pH of the crop, gizzard, duodenum, jejunum, ileum, caeca and colon contents was measured using a Beckman Coulter 34 pH meter (Model PHI 34, Beckman Instruments, Fullerton, CA). Total ileal and caecal contents were collected to a plastic tray and a portion was added into plastic centrifuge tubes and stored at -20˚C for the analysis of SCFA. The rest of the ileal content was put into a plastic snap-cap vial. Another two birds per cage were used to collect digestive tract size, content, and organ data. The digestive tract was detached from the bird carcass and then sectioned into the crop, proventriculus, gizzard, duodenum, jejunum, ileum, caeca and colon; the liver, spleen and pancreas were removed and weighed. Full and empty weights of all sections and the length of each intestinal section were recorded. The content weight of each section was determined by subtracting the empty weight from the full weight. Relative tissue weights and lengths were calculated based on individual bird weight. Total ileal content was collected into the same plastic snap-cap vial (pooled from all the birds in a cage) and centrifuged for 5 min at 17013 × g using a Beckman microfuge (Model E 348720, Beckman Instruments, INC, Palo Alto, CA). The viscosity of ileal supernatant was measured using a Brookfield cone-plate digital viscometer (Model LVDV-III, Brookfield Engineering Labs, INC, Stoughton, MA 02072), which was maintained at 40˚C (40 rpm; shear rate 300 s$^{-1}$). The rest of the ileal supernatant was stored at -80˚C for β-glucan molecular weight analysis.

## Experiment 2

**Birds and housing.** A total of 2376 male and female (Ross × Ross 308) broiler chickens were obtained from a commercial hatchery on the day of hatch and randomly placed in 36 litter (straw) floor pens (2.3 m × 2.0 m) in nine environmentally controlled rooms with an estimated trial end density of 31 kg/m$^2$. Each room contained four pens randomly assigned to the four treatments; each treatment was replicated nine times. Each pen (66 birds per pen) contained a tube feeder and a height-adjustable nipple drinker (six Lubing nipples). The room temperature was 33˚C at the chick placement and was gradually reduced to 21˚C by d 25. Day length was gradually reduced from 23 h at d 0 to 17 h at d 12, and the light intensity was set to 20 lux at the start and gradually decreased to 10 lux by d 10. Birds were given feed and water *ad-libitum*.

**Experimental diets.** The experimental diets were designed according to a 2 × 2 factorial arrangement. The two main factors were BGase (Econase GT 200 P from ABVista, Wiltshire, UK) and medication (same antibiotic and anti-coccidial drug used in Experiment 1). Beta-glucanase levels 0 and 0.1% (BGase activity of 0 and 200,000 BU/kg, respectively), and with or without medication were applied for the experimental diets. CDC Fibar was used as the HB cultivar for the experiment. The diets were formulated by adhering to Ross 308 broiler

**Table 1. Ingredients and calculated nutrient levels (%) of experimental diets.**

| Ingredient | Experiment 1 | Experiment 2 | |
|---|---|---|---|
| | | Starter | Grower |
| Hulless barley | 60.00 | 59.09 | 60.00 |
| Wheat | 4.46 | 0.00 | 4.55 |
| Soybean meal | 26.93 | 32.97 | 26.99 |
| Canola oil | 4.07 | 3.29 | 4.13 |
| Monocalcium phosphate | 1.20 | 1.40 | 1.20 |
| Limestone | 1.52 | 1.64 | 1.52 |
| Sodium chloride | 0.38 | 0.43 | 0.38 |
| Vitamin-mineral broiler premix[1] | 0.50 | 0.50 | 0.50 |
| Choline chloride | 0.10 | 0.10 | 0.10 |
| DL-Methionine | 0.27 | 0.30 | 0.27 |
| L-Threonine | 0.05 | 0.07 | 0.05 |
| L-Lysine HCl | 0.22 | 0.21 | 0.22 |
| **Nutrient, calculated** | | | |
| AME (kcal/kg) | 3100 | 3000 | 3100 |
| Crude protein | 21.24 | 23.46 | 21.24 |
| Crude fat | 5.57 | 4.74 | 5.57 |
| Calcium | 0.87 | 0.96 | 0.87 |
| Chloride | 0.36 | 0.38 | 0.36 |
| Non-phytate phosphorous | 0.44 | 0.48 | 0.44 |
| Potassium | 0.83 | 0.92 | 0.83 |
| Sodium | 0.18 | 0.20 | 0.18 |
| Digestible arginine | 1.35 | 1.50 | 1.35 |
| Digestible isoleucine | 0.81 | 0.90 | 0.81 |
| Digestible leucine | 1.47 | 1.61 | 1.47 |
| Digestible lysine | 1.15 | 1.28 | 1.15 |
| Digestible methionine | 0.54 | 0.60 | 0.54 |
| Digestible methionine and cysteine | 0.87 | 0.95 | 0.87 |
| Digestible threonine | 0.77 | 0.86 | 0.77 |
| Digestible tryptophan | 0.24 | 0.27 | 0.24 |
| Digestible valine | 0.87 | 0.96 | 0.87 |

[1]Vitamin-mineral premix provided the following per kilogram of complete diet: vitamin A, 11,000 IU; vitamin $D_3$, 2,200 IU; vitamin E, 30 IU; menadione, 2 mg; thiamine, 1.5 mg; riboflavin, 6 mg; pyridoxine, 4 mg; vitamin $B_{12}$, 0.02 mg; niacin, 60 mg; pantothenic acid, 10 mg; folic acid, 0.6 mg; biotin 0.15 mg; copper, 10 mg; iron, 80 mg; manganese 80 mg; iodine, 0.8 mg; zinc, 80 mg; selenium, 0.3 mg; calcium carbonate 500 mg; ethoxyquin 0.63 mg; wheat middlings 3773 mg.

nutrition specifications [25], and the ingredient composition and calculated nutrient levels are shown in Table 1. The starter diets (d 0–11) were fed in crumble form, and grower diets (d 11–33) were given initially in crumble form and then switched to a pellet form. The conditions used during feed processing, including pelleting temperature and the measured enzyme activity, were similar to Experiment 1.

**Coccidiosis vaccination.** In Experiment 2, all the birds were vaccinated with the Coccivac B-52 live vaccine (Merck Animal Health; 1.3× recommended dose). The vaccination was completed at d 5 to facilitate uniform intake of coccidian oocysts by the birds. The vaccine contains oocysts of *Eimeria acervulina*, *E. mivatis*, *E. maxima* and *E. tenella*. The vaccine was sprayed on feed located in a cardboard egg tray and into water placed in an ice cube tray. A 30 cm wide Kraft brown paper strip (Model S-8511S, ULINE Canada, Milton, Ontario, Canada) was

placed under the full length of the nipple drinker line in each pen before vaccination to facilitate oocyst ingestion by the birds. In addition, 60% of relative humidity was maintained in the rooms, to facilitate oocyst cycling. Feeders and drinkers were raised in each pen before vaccination and were put-down once the birds consumed the vaccine containing feed and water.

**Rearing performance data collection.** Body weight and feed intake were measured on a pen basis at d 11, 22 and 32. The examination of bird behavior and health and the humane endpoints were similar to Experiment 1. The mortality was recorded daily, and the bird carcasses were sent to Prairie Diagnostic Services for necropsy.

**Sample collection.** A total of four birds per pen were euthanized at two collection points (d 11 and 33) by intravenous administering T-61 (Merck animal health, Kirkland, Quebec, Canada), and the individual bird weights were recorded. Two birds per pen in each collection were used to take the pH measurements and collect ileal and caecal contents for SCFA analysis as described in Experiment 1. Two 1 cm samples of mid-ileum were sectioned (before taking samples for SCFA analysis), placed in 10% neutral buffered formalin, and stored at room temperature until histo-morphology evaluation. Two birds per pen were used to collect relative digestive tract morphology data at each collection according to the same procedure mentioned under Experiment 1. The viscosity of ileal supernatant was measured using one bird per pen.

## Nutritional analysis

The ingredients (HB and wheat) were ground using a Retsch laboratory mill (Retsch ZM 200, Germany) and analyzed for total starch, CP, fat, ash, moisture and fibre following AOAC, AACC and ICC standard methods [26–28]. Ingredients were analyzed for total starch using the AOAC method 996.11 and the AACC method 76–13.01 using a Megazyme kit (Total starch assay procedure, Amyloglucosidase/α-amylase method, Megazyme International Ireland Ltd., Bray Business Park, Bray, Co. Wicklow, Ireland). Nitrogen was analyzed using a Leco nitrogen analyzer (Model Leco-FP-528L, Leco Corporation, St. Joseph, MA, USA), and 6.25 was the N to CP conversion factor. Ether extraction was completed using Goldfish Extraction Apparatus (Labconco model 35001; Labconco, Kansas, MO, USA) following the AOAC method 920.39 to determine fat content. Ash content was analyzed according to the AOAC method 942.05 using a muffle oven (Model Lindberg/Blue BF51842C, Asheville, NC 28804, USA). Moisture was analyzed using the AOAC method 930.15. The insoluble dietary fibre and soluble dietary fibre analysis was completed using a Megazyme kit (Total dietary fibre assay procedure, Megazyme International Ireland Ltd., Bray Business Park, Bray, Co. Wicklow, Ireland) according to the AOAC method 991.43 and the AACC method 32–07.01. Total dietary fibre was obtained by adding insoluble and soluble dietary fibre. Beta-glucan was analyzed using a Megazyme analysis kit (Mixed-linkage beta-glucan assay procedure/McCleary method, Megazyme International Ireland Ltd., Bray Business Park, Bray, Co. Wicklow, Ireland) according to the AOAC Method 995.16, AACC Method 32–23, and ICC Standard Method No. 168. In addition, diets were analyzed for β-glucanase (EC 3.2.1.6) and xylanase activity (EC 3.2.1.8) according to the AB Vista methods of ESC Standard Analytical Methods SAM042-01 and SAM038, respectively (ABVista, Wiltshire, UK).

## Beta-glucan molecular weight

Ileal supernatant samples were boiled for 15 min and centrifuged at 17,013 × g for 10 min using a Beckman microfuge (Model E348720, Beckmann instruments, INC, Palo Alto, CA). The sample was then analyzed for β-glucan molecular weight using size exclusion chromatography and calcofluor post-column derivatization [29]. The two columns used for HPLC were Shodex OHpak SB-806M with OHpak SB-G column guard and a Waters Ultrahydrogel linear column. The mobile phase was 0.1M Tris buffer (pH = 8). Beta-glucan peak molecular weight

(Mp), weighted average molecular weight (Mw), and the maximum molecular weight for the smallest 10% β-glucan molecules (MW-10%) of each sample were noted. Peak molecular weight is the molecular weight of the highest β-glucan fraction, and the weighted average molecular weight is the average of the molecular weights of all β-glucan, emphasizing the weight fraction of each molecule [29].

## Short chain fatty acids analysis

Short chain fatty acids were analyzed in triplicate by [30] with minor changes. The internal standard for the analysis was made up of 20 ml of 25% phosphoric acid, 300 μl of isocaproic acid, and deionized water. Three hundred microliters of acetic acid, 200 μl of propionic acid, 100 μl of butyric acid, and 50 μl of isobutyric, isovaleric, valeric, caproic and lactic acids were used to make the standard solution. The digesta was thawed and mixed with 25% phosphoric acid at 1:1 and kept at room temperature for 10 min with occasional shaking. It was then centrifuged at $12,500 \times g$ for 10 min. The supernatant (1 ml) was mixed with 1 ml of the internal standard and centrifuged at $12,500 \times g$ for 10 min. It was filtered using a 0.45-micron nylon filter, and the filtrate was placed in a GC autosampler vial and injected into a Zebron Capillary Gas Chromatography column (length 30m, internal diameter 0.25 mm, film thickness 0.25 μm; (Zebron[TM]ZB-FFAP, Phenomenex, Torrance, CA). The SCFA analysis was completed using the Thermo Scientific Gas Chromatography system (Model Trace 1310, Milan, Italy).

## Histomorphology of gastro-intestinal wall

Ileal tissue samples were cut into two longitudinal sections and embedded in paraffin. Two slides were made from each sample to obtain ileal morphology measurements (Hematoxylin and Eosin stain) and goblet cell categorization (Alcian Blue/ Periodic Acid-Schiff stain). An Optika B-290TB digital microscope (Bergamo, Italy) was used to observe slides, and an HDCE-X3 digital camera with Optika Vision Lite software was used to capture the images. Well-oriented 8–10 villi and crypts per section were used to measure villi length, width, and crypt depth. Villi length was considered as the length from the tip of a villus to the villus-crypt junction. The villi width was measured at the middle of the villus height. The depth of the invagination between adjacent villi was considered as the crypt depth. Goblet cells were counted around the perimeter of 8–10 well-oriented villi per section, and the three categories of goblet cells were identified, acidic mucin-producing (stained in blue), neutral mucin-producing (stained in magenta) and mixed mucin-producing (stained in purple) [31].

## Statistical analysis

Data were analyzed using the Proc Mixed model of SAS 9.4 [32]. Both experiments were randomized complete block designs, and the battery cage level and room were considered as blocks for Experiments 1 and 2, respectively. Treatments were replicated 10 times in Experiment 1 (battery cages equally distributed in two levels) and nine times in Experiment 2 (one pen in nine different rooms). Differences were considered significant when $P \leq 0.05$. Data were checked for normality and analyzed using 2-way ANOVA. Tukey-Kramer test was used to detect significant differences between means.

## Results

### Experiment 1

**Ingredient nutrient composition.** Total dietary fibre, insoluble dietary fibre, soluble dietary fibre and total β-glucan in HB were 29.0, 19.6, 9.6 and 8.70%, respectively, and the same

**Table 2. Effects of diet medication and β-glucanase on β-glucan molecular weight in ileal content of broiler chickens.**

| Medication | β-glucanase (%) | Molecular weight (g/mol) | | | | | | | | |
|---|---|---|---|---|---|---|---|---|---|---|
| | | Experiment 1 | | | Experiment 2 | | | | | |
| | | d 28 | | | d 11 | | | d 33 | | |
| | | Mp[1] | Mw | MW-10% | Mp | Mw | MW-10% | Mp | Mw | MW-10% |
| without | 0 | 19799[a] | 36199[a] | 6096 | 78293[a] | 80971 | 33322[a] | 65176[a] | 69508[a] | 29025[a] |
| | 0.1 | 7793[b] | 8434[c] | 1955 | 24568[c] | 63835 | 7250[b] | 16985[c] | 48316[b] | 7074[c] |
| with | 0 | 16824[a] | 19119[b] | 5326 | 54475[b] | 59002 | 26065[a] | 40595[b] | 49017[b] | 13586[b] |
| | 0.1 | 10401[b] | 9929[c] | 2201 | 27677[c] | 61898 | 10586[b] | 22144[c] | 60641[a] | 8157[c] |
| SEM[2] | | 1148.1 | 2513.9 | 509.2 | 5982.7 | 3537.4 | 2717.0 | 4481.7 | 2258.9 | 1890.1 |
| Main effects | | | | | | | | | | |
| *Medication* | | | | | | | | | | |
| without | | 13796 | 22317 | 4025 | 51431 | 72403 | 20286 | 41080 | 58912 | 18049 |
| with | | 13612 | 14524 | 3763 | 41076 | 60450 | 18325 | 31370 | 54829 | 10871 |
| *β-glucanase (%)* | | | | | | | | | | |
| 0 | | 18311 | 27659 | 5711[a] | 66384 | 69986 | 29694 | 52885 | 59263 | 21305 |
| 0.1 | | 9096 | 9181 | 2078[b] | 26122 | 62867 | 8918 | 19565 | 54479 | 7615 |
| *Probability* | | | | | | | | | | |
| Medication | | 0.86 | 0.001 | 0.70 | 0.08 | 0.06 | 0.39 | 0.04 | 0.16 | < .0001 |
| β-glucanase | | < .0001 | < .0001 | < .0001 | < .0001 | 0.21 | < .0001 | < .0001 | 0.10 | < .0001 |
| Medication × β-glucanase | | 0.01 | 0.0004 | 0.45 | 0.03 | 0.09 | 0.03 | 0.004 | < .0001 | < .0001 |

[a-c]Means within a main effect or interaction not sharing a common superscript are significantly different ($P \leq 0.05$).

[1]Mp—peak molecular weight; Mw—weighted average molecular weight; MW-10%—The maximum molecular weight for the smallest 10% molecules.

[2]SEM—pooled standard error of mean (d 28, n = 6 cages per treatment; d 11 and 33, n = 6 birds per treatment).

fractions were 15.2, 13.7, 1.6 and 0.68%, respectively for wheat. The content of total starch, CP, fat and ash were measured as 49.7, 16.2, 2.4 and 2.4%, respectively, in HB, and as 64.1, 15.0, 1.2 and 1.9% in wheat.

**Beta-glucan molecular weight.** Interactions between BGase and medication were significant for Mp and Mw but not for MW-10% (Table 2) For Mp, BGase decreased values without and with medication, but mean separation failed to confirm an interaction as values were not affected by medication regardless of enzyme use. The interaction for Mw again demonstrated a lowering effect of BGase resulting in similar values without and with medication. Medication reduced Mw in the absence of BGase but had no effect with BGase. Medication did not affect MW-10% while BGase reduced its value.

**The viscosity of ileal supernatant.** Ileal digesta viscosity was not affected by medication in Experiment 1, but was reduced with the use of BGase (Table 3).

**Short chain fatty acids and gastro-intestinal pH.** Ileal digesta SCFA levels and molar percentages were not affected by dietary treatments, except for caproic acid concentration, where values were lower with BGase supplementation (Table 4). Similarly, caecal digesta SCFA concentrations and molar percentages were also not affected by treatment (Table 5). Noteworthy, the interaction between medication and BGase tended to be significant (P = 0.06–0.09) for the concentrations of total and individual SCFA. In all cases, levels tended to decrease with enzyme use in the non-medicated diets and increase with enzyme use in the medicated diets.

Except for the duodenum, medication, BGase, and their interactions did not affect the digestive tract pH (Table 6). The enzyme use increased duodenal pH from 6.08 to 6.20.

**Table 3. Effects of diet medication and β-glucanase on the viscosity of ileal supernatant in broiler chickens.**

| Medication | β-glucanase (%) | Viscosity (cP) | | |
|---|---|---|---|---|
| | | Experiment 1 | Experiment 2 | |
| | | d 28 | d 11 | d 33 |
| without | 0 | 4.72 | 9.73[a] | 3.98 |
| | 0.1 | 3.33 | 3.53[b] | 2.30 |
| with | 0 | 4.16 | 6.04[ab] | 4.61 |
| | 0.1 | 3.38 | 4.13[b] | 2.80 |
| SEM[1] | | 0.147 | 0.674 | 0.250 |
| Main effects | | | | |
| *Medication* | | | | |
| without | | 4.02 | 6.63 | 3.14 |
| with | | 3.77 | 5.08 | 3.70 |
| *β-glucanase (%)* | | | | |
| 0 | | 4.44[a] | 7.89 | 4.29[a] |
| 0.1 | | 3.35[b] | 3.83 | 2.55[b] |
| *Probability* | | | | |
| Medication | | 0.25 | 0.11 | 0.17 |
| β-glucanase | | < .0001 | 0.0005 | 0.0002 |
| Medication × β-glucanase | | 0.16 | 0.03 | 0.86 |

[a-b]Means within a main effect or interaction not sharing a common superscript are significantly different ($P \leq 0.05$).

[1]SEM—pooled standard error of mean (d 28; n = 10 cages per treatment/ d 11; n = 6 birds per treatment/ d 33; n = 9 birds per treatment).

**Table 4. Effects of diet medication and β-glucanase on ileal digesta short chain fatty acids of broiler chickens at 28 days of age (Experiment 1).**

| Medication | BGase[1] (%) | SCFA μmol/g of wet ileal content | | | | | | | | | Molar percentage of total SCFA | | | | | | | |
|---|---|---|---|---|---|---|---|---|---|---|---|---|---|---|---|---|---|---|
| | | Total | Ace | Pro | But | Isob | Val | Isov | Cap | Lac | Ace | Pro | But | Isob | Isov | Val | Cap | Lac |
| without | 0 | 165.8 | 61.8 | 22.2 | 10.6 | 2.7 | 3.3 | 2.9 | 1.3 | 60.6 | 37.5 | 13.1 | 6.4 | 1.6 | 1.7 | 1.9 | 0.7 | 36.6 |
| | 0.1 | 157.2 | 59.1 | 20.8 | 10.3 | 2.9 | 2.6 | 2.2 | 1.0 | 58.0 | 37.6 | 13.3 | 6.5 | 1.8 | 1.4 | 1.6 | 0.6 | 36.9 |
| with | 0 | 173.5 | 66.4 | 23.4 | 10.8 | 2.5 | 2.7 | 2.9 | 1.5 | 63.0 | 38.3 | 13.2 | 6.3 | 1.4 | 1.6 | 1.5 | 0.8 | 36.5 |
| | 0.1 | 156.9 | 59.1 | 21.8 | 10.3 | 2.4 | 2.6 | 2.6 | 1.2 | 56.5 | 37.6 | 14.0 | 6.6 | 1.4 | 1.6 | 1.6 | 0.8 | 36.1 |
| SEM[2] | | 4.51 | 1.66 | 0.75 | 0.31 | 0.18 | 0.17 | 0.17 | 0.07 | 1.60 | 0.23 | 0.28 | 0.09 | 0.10 | 0.09 | 0.08 | 0.09 | 0.24 |
| Main effects | | | | | | | | | | | | | | | | | | |
| *Medication* | | | | | | | | | | | | | | | | | | |
| Without | | 161.5 | 60.5 | 21.5 | 10.4 | 2.8 | 2.9 | 2.6 | 1.1 | 59.3 | 37.6 | 13.2 | 6.5 | 1.7 | 1.5 | 1.7 | 0.7 | 36.7 |
| With | | 165.2 | 62.7 | 22.6 | 10.5 | 2.4 | 2.6 | 2.8 | 1.3 | 59.8 | 38.0 | 13.6 | 6.4 | 1.4 | 1.6 | 1.5 | 0.8 | 36.3 |
| *BGase (%)* | | | | | | | | | | | | | | | | | | |
| 0 | | 169.6 | 64.1 | 22.8 | 10.7 | 2.6 | 3.0 | 2.9 | 1.4[a] | 61.8 | 37.9 | 13.2 | 6.3 | 1.5 | 1.7 | 1.7 | 0.8 | 36.6 |
| 0.1 | | 157.0 | 59.1 | 21.3 | 10.3 | 2.6 | 2.6 | 2.4 | 1.1[b] | 57.2 | 37.6 | 13.6 | 6.6 | 1.6 | 1.5 | 1.6 | 0.7 | 36.5 |
| *Probability (%)* | | | | | | | | | | | | | | | | | | |
| Medication | | 0.66 | 0.46 | 0.41 | 0.86 | 0.31 | 0.38 | 0.55 | 0.10 | 0.87 | 0.38 | 0.48 | 0.80 | 0.13 | 0.78 | 0.25 | 0.08 | 0.38 |
| BGase | | 0.13 | 0.11 | 0.28 | 0.53 | 0.94 | 0.28 | 0.11 | 0.02 | 0.13 | 0.57 | 0.45 | 0.27 | 0.55 | 0.34 | 0.55 | 0.10 | 0.85 |
| Medication × BGase | | 0.63 | 0.45 | 0.94 | 0.90 | 0.67 | 0.36 | 0.52 | 0.73 | 0.51 | 0.40 | 0.59 | 0.61 | 0.57 | 0.34 | 0.23 | 0.35 | 0.47 |

[a-d]Means within a main effect or interaction not sharing a common superscript are significantly different ($P \leq 0.05$).

[1]BGase—β-glucanase; SCFA—short chain fatty acids; Ace—Acetic acid; Pro—Propionic acid; But—Butyric acid; Isob—Isobutyric acid; Val—Valeric acid; Isov—Isovaleric acid; Cap—Caproic acid.

[2]SEM—pooled standard error of mean (n = 20 birds per treatment).

**Table 5. Effects of diet medication and β-glucanase on caecal short chain fatty acids of broiler chickens aged 28 days (Experiment 1).**

| Medication | BGase[1] (%) | SCFA μmol/g of wet caecal content | | | | | | | | Molar percentage of total SCFA | | | | | | |
|---|---|---|---|---|---|---|---|---|---|---|---|---|---|---|---|---|
| | | Total | Ace | Pro | But | Isob | Val | Isov | Cap | Ace | Pro | But | Isob | Val | Isov | Cap |
| without | 0 | 284.2 | 166.6 | 58.5 | 28.0 | 9.9 | 8.6 | 8.6 | 3.7 | 58.7 | 20.5 | 9.8 | 3.5 | 3.0 | 3.0 | 1.3 |
| | 0.1 | 273.9 | 161.7 | 56.5 | 27.0 | 8.4 | 8.3 | 8.3 | 3.5 | 59.0 | 20.6 | 9.9 | 3.0 | 3.0 | 3.0 | 1.3 |
| with | 0 | 267.5 | 158.0 | 55.2 | 26.2 | 8.2 | 8.1 | 8.1 | 3.5 | 59.0 | 20.6 | 9.8 | 3.0 | 3.0 | 3.0 | 1.3 |
| | 0.1 | 310.3 | 183.1 | 64.0 | 30.6 | 9.5 | 9.3 | 9.4 | 4.0 | 58.9 | 20.6 | 9.8 | 3.0 | 3.0 | 3.0 | 1.3 |
| SEM[2] | | 7.59 | 4.49 | 1.60 | 0.74 | 0.35 | 0.23 | 0.23 | 0.10 | 0.23 | 0.28 | 0.09 | 0.10 | 0.08 | 0.09 | 0.03 |
| Main effects | | | | | | | | | | | | | | | | |
| *Medication* | | | | | | | | | | | | | | | | |
| without | | 279.0 | 164.1 | 57.5 | 27.5 | 9.1 | 8.4 | 8.5 | 3.6 | 58.8 | 20.5 | 9.8 | 3.3 | 3.0 | 3.0 | 1.3 |
| with | | 288.9 | 170.5 | 59.6 | 28.4 | 8.8 | 8.7 | 8.8 | 3.7 | 59.0 | 20.6 | 9.8 | 3.0 | 3.0 | 3.0 | 1.3 |
| *BGase (%)* | | | | | | | | | | | | | | | | |
| 0 | | 275.8 | 162.3 | 56.8 | 27.1 | 9.0 | 8.3 | 8.4 | 3.6 | 58.8 | 20.5 | 9.8 | 3.3 | 3.0 | 3.0 | 1.3 |
| 0.1 | | 292.1 | 172.4 | 60.2 | 28.8 | 8.9 | 8.8 | 8.9 | 3.8 | 59.0 | 20.6 | 9.9 | 3.0 | 3.0 | 3.0 | 1.3 |
| *Probability (%)* | | | | | | | | | | | | | | | | |
| Medication | | 0.50 | 0.46 | 0.50 | 0.53 | 0.69 | 0.51 | 0.49 | 0.48 | 0.57 | 0.57 | 0.90 | 0.30 | 0.62 | 0.49 | 0.47 |
| BGase | | 0.27 | 0.25 | 0.27 | 0.23 | 0.85 | 0.31 | 0.30 | 0.30 | 0.57 | 0.65 | 0.48 | 0.27 | 0.94 | 0.92 | 0.95 |
| Medication × BGase | | 0.07 | 0.09 | 0.08 | 0.06 | 0.06 | 0.08 | 0.08 | 0.08 | 0.47 | 0.71 | 0.99 | 0.28 | 0.61 | 0.76 | 0.84 |

[a-d]Means within a main effect or interaction not sharing a common superscript are significantly different ($P \leq 0.05$).

[1]BGase—β-glucanase; SCFA—short chain fatty acids; Ace—Acetic acid; Pro—Propionic acid; But—Butyric acid; Isob—Isobutyric acid; Val—Valeric acid; Isov—Isovaleric acid; Cap—Caproic acid.

[2]SEM—pooled standard error of mean (n = 20 birds per treatment).

**Table 6. Effects of diet medication and β-glucanase on gastro-intestinal pH of broiler chickens at day 28 (Experiment 1).**

| Medication | β-glucanase (%) | Crop | Gizzard | Duodenum | Jejunum | Ileum | Caeca | Colon |
|---|---|---|---|---|---|---|---|---|
| without | 0 | 5.29 | 3.54 | 6.05 | 5.99 | 7.08 | 6.02 | 6.92 |
| | 0.1 | 5.23 | 3.26 | 6.19 | 6.01 | 7.26 | 6.04 | 7.17 |
| with | 0 | 5.43 | 3.23 | 6.10 | 5.96 | 7.25 | 5.90 | 7.08 |
| | 0.1 | 5.20 | 3.17 | 6.21 | 6.05 | 7.27 | 5.93 | 7.13 |
| SEM[1] | | 0.070 | 0.071 | 0.027 | 0.024 | 0.048 | 0.055 | 0.067 |
| Main effects | | | | | | | | |
| *Medication* | | | | | | | | |
| Without | | 5.26 | 3.40 | 6.12 | 5.99 | 7.17 | 6.03 | 7.04 |
| With | | 5.31 | 3.20 | 6.16 | 6.00 | 7.26 | 5.91 | 7.11 |
| *β-glucanase (%)* | | | | | | | | |
| 0 | | 5.36 | 3.39 | 6.08[b] | 5.97 | 7.16 | 5.96 | 7.00 |
| 0.1 | | 5.21 | 3.22 | 6.20[a] | 6.03 | 7.26 | 5.98 | 7.15 |
| *Probability* | | | | | | | | |
| Medication | | 0.70 | 0.15 | 0.46 | 0.89 | 0.25 | 0.29 | 0.61 |
| β-glucanase | | 0.29 | 0.21 | 0.01 | 0.16 | 0.20 | 0.82 | 0.22 |
| Medication × β-glucanase | | 0.55 | 0.41 | 0.80 | 0.40 | 0.29 | 0.94 | 0.43 |

[a-b]Means within a main effect or interaction not sharing a common superscript are significantly different ($P \leq 0.05$).

[1]SEM—pooled standard error of mean (n = 20 birds per treatment).

**Table 7. Effects of diet medication and β-glucanase on gastro-intestinal tissue weights and lengths (proportional to body weight) of broiler chickens at d 28 (Experiment 1).**

| Medication | BGase[1] (%) | Empty weight (%) | | | | | | | | | Length (cm/100g) | | | | | |
|---|---|---|---|---|---|---|---|---|---|---|---|---|---|---|---|---|
| | | Crop | Proven | Gizzard | Duo | Jejunum | Ileum | SI | Caeca | Colon | Duo | Jejunum | Ileum | SI | Caeca | Colon |
| without | 0 | 0.34[a] | 0.38 | 1.20 | 0.73 | 1.37 | 1.00 | 3.08 | 0.36 | 0.17 | 1.73 | 4.22 | 4.18 | 10.07 | 1.67 | 0.41 |
| | 0.1 | 0.29[b] | 0.38 | 1.32 | 0.73 | 1.30 | 0.91 | 2.94 | 0.37 | 0.14 | 1.75 | 4.01 | 4.11 | 9.87 | 1.69 | 0.39 |
| with | 0 | 0.30[ab] | 0.43 | 1.31 | 0.71 | 1.31 | 0.97 | 2.99 | 0.36 | 0.15 | 1.80 | 4.24 | 4.35 | 10.39 | 1.73 | 0.42 |
| | 0.1 | 0.31[ab] | 0.38 | 1.33 | 0.74 | 1.28 | 0.93 | 2.94 | 0.37 | 0.15 | 1.79 | 4.23 | 4.29 | 10.28 | 1.68 | 0.42 |
| SEM[2] | | 0.006 | 0.009 | 0.020 | 0.008 | 0.018 | 0.012 | 0.030 | 0.009 | 0.003 | 0.023 | 0.056 | 0.059 | 0.118 | 0.026 | 0.007 |
| Main effects | | | | | | | | | | | | | | | | |
| *Medication* | | | | | | | | | | | | | | | | |
| without | | 0.32 | 0.38 | 1.26 | 0.73 | 1.33 | 0.96 | 3.01 | 0.36 | 0.16 | 1.74 | 4.12 | 4.15 | 9.97 | 1.68 | 0.40 |
| with | | 0.30 | 0.40 | 1.32 | 0.73 | 1.30 | 0.95 | 2.97 | 0.37 | 0.15 | 1.79 | 4.23 | 4.32 | 10.33 | 1.71 | 0.42 |
| *BGase (%)* | | | | | | | | | | | | | | | | |
| 0 | | 0.32 | 0.41 | 1.25 | 0.72 | 1.34 | 0.98[a] | 3.04 | 0.36 | 0.16[a] | 1.76 | 4.23 | 4.27 | 10.23 | 1.70 | 0.42 |
| 0.1 | | 0.30 | 0.38 | 1.32 | 0.74 | 1.29 | 0.92[b] | 2.94 | 0.37 | 0.15[b] | 1.77 | 4.12 | 4.20 | 10.07 | 1.68 | 0.41 |
| *Probability* | | | | | | | | | | | | | | | | |
| Medication | | 0.36 | 0.18 | 0.10 | 0.83 | 0.34 | 0.61 | 0.45 | 0.84 | 0.58 | 0.21 | 0.29 | 0.13 | 0.11 | 0.61 | 0.16 |
| BGase | | 0.29 | 0.10 | 0.07 | 0.30 | 0.14 | 0.005 | 0.12 | 0.41 | 0.01 | 0.83 | 0.32 | 0.56 | 0.49 | 0.75 | 0.44 |
| Medication × BGase | | 0.007 | 0.13 | 0.18 | 0.47 | 0.57 | 0.31 | 0.40 | 0.98 | 0.08 | 0.75 | 0.35 | 0.97 | 0.82 | 0.48 | 0.64 |

[a-b]Means within a main effect or interaction not sharing a common superscript are significantly different ($P \leq 0.05$).

[1]BGase—β-glucanase; Proven—proventriculus; Duo—duodenum; SI—small intestine.

[2]SEM—pooled standard error of mean (n = 20 birds per treatment).

**Gastro-intestinal tract morphology.** Interactions were not found between BGase and medication for empty weights and lengths of the digestive tract sections, except for crop weight (Table 7). Crop weight was lower with enzyme use when the birds were fed a non-medicated diet, but the enzyme had no effect when the diets were medicated. However, the absence of medication effect based on the mean separation failed to show the interaction. Both ileum and colon weights were lower when the enzyme was fed. Crop content weight was higher, and duodenal and ileal content weights were lower when 0.1% BGase was fed (Table 8).

**Measurements of the contents of the digestive tract.** Interactions between BGase and medication were found for the content weights of the gizzard, jejunum and small intestine. Medication increased the gizzard content weight when the diets did not contain BGase. Beta-glucanase resulted in lower jejunal and small intestinal content weights in the absence of dietary antibiotics but had no effect when the medication was used. However, the interactions were not demonstrated based on mean separation due to the absence of medication or BGase effect on these content weights.

**Body weight gain, feed intake and feed to gain ratio.** Interactions between medication and BGase were significant or nearly significant for body weight gain and feed intake from 0–7 d, 7–14 d ($P = 0.06$) and 0–28 d ($P = 0.06$–$0.07$), and feed to gain ratio (F:G) from 0–7 d (Table 9). Body weight gain and feed intake followed a similar response to treatments. In birds fed diets without medication, the addition of BGase reduced 0–7 d gain and feed intake and tended to reduce 7–14 d gain and feed consumption. However, in those fed diets with medication, enzyme either did not affect (0–7 d) or increased (7–14 d) these response criteria. For the 0–7 d F:G ratio interaction, enzyme decreased and increased feed efficiency in unmedicated diets and medicated diets, respectively. The total mortality of the study was 3.8%, and HB or BGase did not affect the mortality.

**Table 8. Effects of diet medication and β-glucanase on gastro-intestinal content and organ weights as a percentage of body weight of broiler chickens at d 28 (Experiment 1).**

| Medication | BGase[1] (%) | Content | | | | | | | | | Weight | | |
|---|---|---|---|---|---|---|---|---|---|---|---|---|---|
| | | Crop | Proven | Gizzard | Duo | Jejunum | Ileum | SI | Caeca | Colon | Liver | Spleen | Pancreas |
| without | 0 | 0.28 | 0.03 | 0.93<sup>b</sup> | 0.09 | 1.03<sup>a</sup> | 1.17 | 2.29<sup>a</sup> | 0.30 | 0.19 | 2.40 | 0.10 | 0.24 |
| | 0.1 | 0.52 | 0.03 | 1.14<sup>b</sup> | 0.07 | 0.74<sup>b</sup> | 0.90 | 1.69<sup>b</sup> | 0.24 | 0.16 | 2.50 | 0.09 | 0.23 |
| with | 0 | 0.33 | 0.11 | 1.53<sup>a</sup> | 0.09 | 0.85<sup>ab</sup> | 1.11 | 2.05<sup>ab</sup> | 0.27 | 0.21 | 2.43 | 0.10 | 0.26 |
| | 0.1 | 0.45 | 0.03 | 1.31<sup>ab</sup> | 0.07 | 0.87<sup>ab</sup> | 1.06 | 2.00<sup>ab</sup> | 0.26 | 0.19 | 2.40 | 0.09 | 0.25 |
| SEM[2] | | 0.066 | 0.017 | 0.058 | 0.005 | 0.028 | 0.035 | 0.056 | 0.014 | 0.009 | 0.029 | 0.003 | 0.005 |
| Main effects | | | | | | | | | | | | | |
| *Medication* | | | | | | | | | | | | | |
| without | | 0.40 | 0.03 | 1.03 | 0.08 | 0.88 | 1.03 | 1.99 | 0.27 | 0.18 | 2.45 | 0.10 | 0.24 |
| with | | 0.39 | 0.07 | 1.42 | 0.08 | 0.86 | 1.09 | 2.02 | 0.26 | 0.20 | 2.41 | 0.09 | 0.25 |
| *BGase (%)* | | | | | | | | | | | | | |
| 0 | | 0.30<sup>b</sup> | 0.07 | 1.23 | 0.09<sup>a</sup> | 0.94 | 1.14<sup>a</sup> | 2.17 | 0.28 | 0.20 | 2.41 | 0.10 | 0.25 |
| 0.1 | | 0.48<sup>a</sup> | 0.03 | 1.22 | 0.07<sup>b</sup> | 0.80 | 0.98<sup>b</sup> | 1.84 | 0.25 | 0.18 | 2.45 | 0.92 | 0.24 |
| *Probability* | | | | | | | | | | | | | |
| Medication | | 0.92 | 0.22 | 0.0005 | 0.60 | 0.63 | 0.43 | 0.74 | 0.77 | 0.14 | 0.50 | 0.72 | 0.05 |
| BGase | | 0.04 | 0.25 | 0.93 | 0.01 | 0.007 | 0.02 | 0.002 | 0.21 | 0.19 | 0.45 | 0.20 | 0.16 |
| Medication × BGase | | 0.56 | 0.21 | 0.04 | 0.90 | 0.002 | 0.11 | 0.01 | 0.39 | 0.74 | 0.22 | 0.74 | 0.82 |

<sup>a-b</sup>Means within a main effect or interaction not sharing a common superscript are significantly different ($P \leq 0.05$).

[1]BGase—β-glucanase; Proven—proventriculus; Duo—duodenum; SI—small intestine.

[2]SEM—pooled standard error of mean (n = 20 birds per treatment).

**Table 9. Effects of diet medication and β-glucanase on body weight gain, feed intake and feed efficiency of broiler chickens (Experiment 1).**

| Medication | β-glucanase (%) | BWG[1] (kg) | | | | | FI (kg) | | | | | F:G | | | | |
|---|---|---|---|---|---|---|---|---|---|---|---|---|---|---|---|---|
| | | d 0–7 | d 7–14 | d 14–21 | d 21–28 | d 0–28 | d 0–7 | d 7–14 | d 14–21 | d 21–28 | d 0–28 | d 0–7 | d 7–14 | d 14–21 | d 21–28 | d 0–28 |
| without | 0 | 0.143<sup>a</sup> | 0.303 | 0.507 | 0.699 | 1.650 | 0.167<sup>a</sup> | 0.421 | 0.729 | 1.055 | 2.371 | 1.17<sup>b</sup> | 1.39 | 1.44 | 1.53 | 1.45 |
| | 0.1 | 0.126<sup>c</sup> | 0.296 | 0.498 | 0.656 | 1.575 | 0.157<sup>b</sup> | 0.399 | 0.705 | 1.004 | 2.265 | 1.26<sup>a</sup> | 1.35 | 1.42 | 1.54 | 1.44 |
| with | 0 | 0.130<sup>bc</sup> | 0.284 | 0.492 | 0.668 | 1.573 | 0.160<sup>ab</sup> | 0.387 | 0.706 | 1.000 | 2.251 | 1.23<sup>a</sup> | 1.36 | 1.44 | 1.50 | 1.43 |
| | 0.1 | 0.135<sup>ab</sup> | 0.301 | 0.494 | 0.677 | 1.607 | 0.160<sup>ab</sup> | 0.409 | 0.695 | 1.012 | 2.275 | 1.19<sup>b</sup> | 1.36 | 1.41 | 1.50 | 1.42 |
| SEM[2] | | 1.562 | 2.966 | 4.564 | 10.050 | 14.222 | 1.172 | 4.887 | 5.856 | 11.406 | 18.375 | 0.008 | 0.011 | 0.009 | 0.014 | 0.007 |
| Main effects | | | | | | | | | | | | | | | | |
| *Medication* | | | | | | | | | | | | | | | | |
| Without | | 0.134 | 0.299 | 0.503 | 0.678 | 1.612 | 0.162 | 0.410 | 0.717 | 1.030 | 2.318 | 1.21 | 1.37 | 1.43 | 1.53 | 1.45 |
| With | | 0.132 | 0.292 | 0.493 | 0.673 | 1.591 | 0.160 | 0.398 | 0.700 | 1.006 | 2.263 | 1.21 | 1.36 | 1.42 | 1.50 | 1.43 |
| *β-glucanase (%)* | | | | | | | | | | | | | | | | |
| 0 | | 0.136 | 0.293 | 0.500 | 0.684 | 1.612 | 0.163 | 0.404 | 0.717 | 1.027 | 2.311 | 1.20 | 1.38 | 1.44 | 1.52 | 1.44 |
| 0.1 | | 0.130 | 0.298 | 0.496 | 0.666 | 1.591 | 0.159 | 0.404 | 0.700 | 1.008 | 2.270 | 1.22 | 1.35 | 1.41 | 1.52 | 1.43 |
| *Probability* | | | | | | | | | | | | | | | | |
| Medication | | 0.36 | 0.21 | 0.32 | 0.79 | 0.43 | 0.35 | 0.17 | 0.15 | 0.29 | 0.12 | 0.70 | 0.55 | 0.69 | 0.21 | 0.12 |
| β-glucanase | | 0.01 | 0.38 | 0.71 | 0.36 | 0.45 | 0.04 | 0.99 | 0.14 | 0.39 | 0.25 | 0.06 | 0.30 | 0.20 | 0.96 | 0.26 |
| Medication × β-glucanase | | < .0001 | 0.06 | 0.54 | 0.17 | 0.06 | 0.02 | 0.06 | 0.55 | 0.17 | 0.07 | < .0001 | 0.44 | 0.85 | 0.90 | 0.85 |

<sup>a-c</sup>Means within a main effect or interaction not sharing a common superscript are significantly different ($P \leq 0.05$).

[1]BWG—body weight gain; FI—feed intake; F:G—feed to gain ratio.

[2]SEM—pooled standard error of mean (n = 10 cages per treatment).

## Experiment 2

**Ingredient nutrient composition.**   Total dietary fibre, insoluble dietary fibre, soluble dietary fibre and total β-glucan were 26.7, 18.9, 7.8 and 8.70% (HB); 14.4, 12.4, 2.0 and 0.64% (wheat), respectively. In addition, total starch, CP, fat and ash were determined to be 53.7, 16.2, 2.8 and 2.4% in HB, and as 62.8, 14.9, 1.2 and 1.7% in wheat, respectively.

**Beta-glucan molecular weight.**   Interactions were found for all molecular weight criteria at both ages (11 and 33 d) except for Mw at 11 d, which was unaffected by medication or BGase (Table 2). Values for Mp and MW-10% followed a similar trend, with enzyme consistently reducing values at both ages, but with the degree of response less in medicated diets when considering Mp. In the absence of the enzyme, medication reduced Mp at both ages and MW-10% on d 33. However, mean separation could not demonstrate an interaction between BGase and medication for MW-10% at d 11 since the medication effect is not evident regardless of BGase use. The interaction for Mw at 33 d was due to enzyme decreasing and increasing Mw for nonmedicated and medicated diets, respectively.

Fig 1A and 1B compare the β-glucan molecular weight of ileal digesta from 11 d broilers fed diets without medication and without and with BGase, respectively. Beta-glucanase increased the proportion of low molecular weight β-glucan, as shown by curve placement relative to the blue line at x-axis point $1e^4$ (a random point that selected to compare the three graphs). Diet medication also increased the proportion of low molecular weight β-glucan compared to the nonmedicated diet, which is contrasted in Fig 1A and 1C. The same BGase and medication effects were observed in the β-glucan molecular weight curves of broiler chickens aged 33 d.

**The viscosity of ileal supernatant.**   At 11 d, an interaction was found between medication and BGase; BGase reduced viscosity without dietary medication (Table 3). However, the interaction between BGase and medication was not clear based on the mean separation as the medication effect is absent despite the use of BGase. In the interaction, the highest viscosity was noted for the treatment without medication or BGase, and the lowest was the treatments with BGase; treatment with medication and without BGase was intermediate. At d 33, BGase decreased viscosity, but there was no medication effect.

**Short chain fatty acids and gastro-intestinal pH.**   To a large extent, dietary treatment did not affect ileal digesta SCFA of 11 d old broilers (Table 10). The exception was a significant interaction between medication and BGase for valeric acid. Without medication, levels of valeric acid decreased with enzyme use, while levels increased with enzyme use when the medication was included in the diet. A similar trend ($P = 0.10$) was noted for isovaleric acid. Levels of caproic acid decreased with enzyme use. Interactions between BGase and medication were found for the molar percentages of valeric, isovaleric ($P = 0.06$), and caproic acids. In diets without medication, BGase did not affect acid concentration. When the medication was used, BGase increased acid levels. Dietary treatment interactions were also noted for the proportional levels of propionic and lactic acids. All mean differences were small and often not significant, but medication decreased propionic acid in BGase containing diets and BGase decreased lactic acid in medicated diets. However, the separation of means was failed to demonstrate the interactions for propionic and lactic acids due to the absence of medication and BGase effects, respectively.

The interactions between medication and BGase use at 11 d were significant for total and individual caecal digesta SCFA (Table 11). The concentrations were higher with 0.1 compared to 0% BGase in the birds given diets without medication. However, BGase did not affect SCFA concentrations in the treatments with medication. Concentrations for birds fed medicated diets were lower than those fed un-medicated diets for the treatments with BGase. The molar percentages of propionic and isobutyric acids were decreased by medication, while enzyme

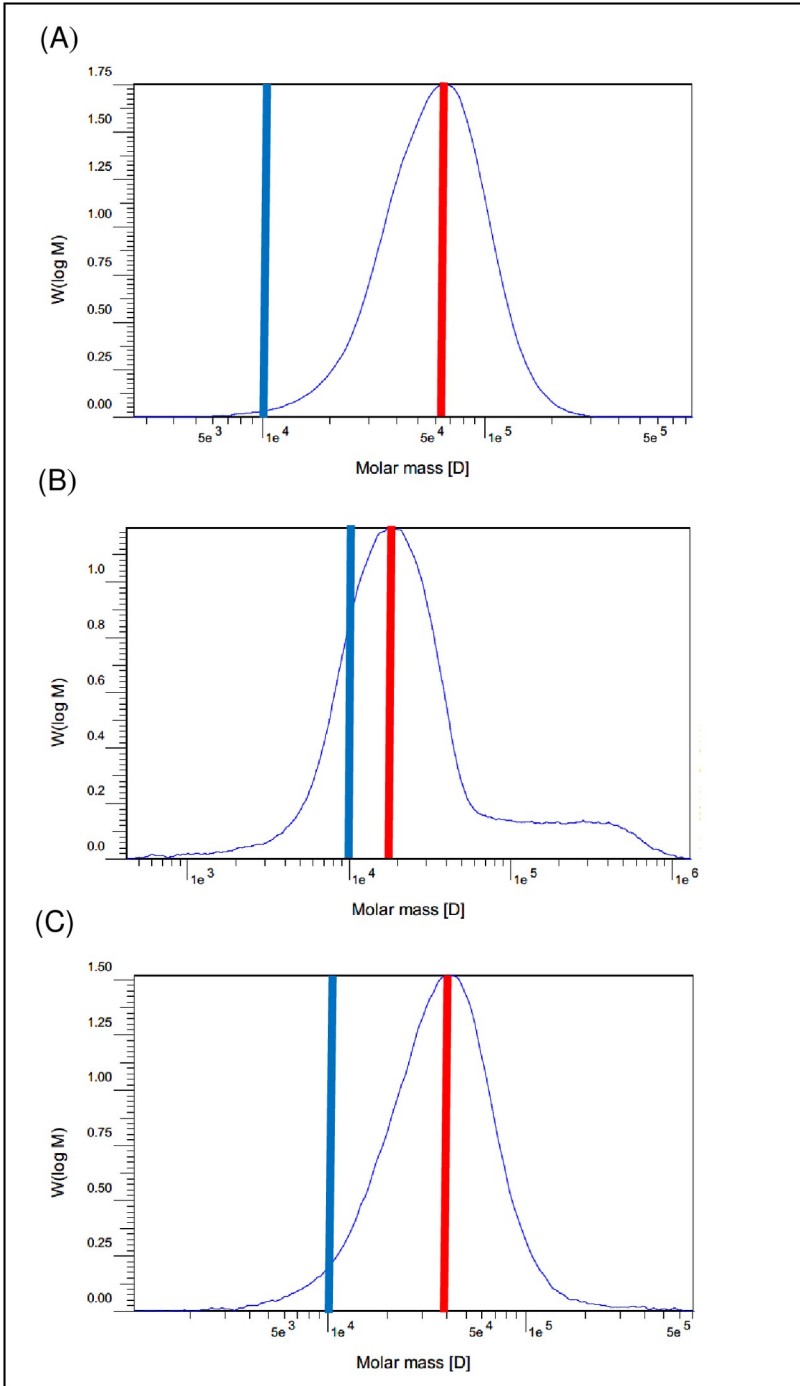

**Fig 1. Beta-glucan molecular weight distribution in soluble ileal digesta from 11 d broilers fed 60% hulless barley diets in Experiment 2.** Blue lines denote point 1e$^4$ on the x-axis and red lines indicate the Mp of the distribution curve. (A) Without medication, 0% β-glucanase (B) Without medication, 0.1% β-glucanase (C) With medication, 0% β-glucanase.

use decreased the proportions of acetic and butyric acids and increased the proportion of valeric acid. The interaction between BGase and medication was significant for the proportional isovaleric levels, with medication decreasing the level in the absence of BGase but having no

**Table 10. Effects of diet medication and β-glucanase on ileal short chain fatty acids of broiler chickens aged 11 days (Experiment 2).**

| Medication | BGase[1] (%) | SCFA μmol/g of wet ileal content | | | | | | | | Molar percentage of total SCFA | | | | | | |
|---|---|---|---|---|---|---|---|---|---|---|---|---|---|---|---|---|
| | | Total | Ace | Pro | But | Val | Isov | Cap | Lac | Ace | Pro | But | Val | Isov | Cap | Lac |
| without | 0 | 125.3 | 48.2 | 18.4 | 8.2 | 2.7[a] | 1.5 | 1.19 | 44.9 | 38.4 | 14.6[ab] | 6.5 | 2.1[a] | 1.2 | 0.9[a] | 35.8[ab] |
| | 0.1 | 122.5 | 47.6 | 18.3 | 8.1 | 1.5[bc] | 1.4 | 0.79 | 44.6 | 38.8 | 14.9[a] | 6.6 | 1.2[ab] | 1.1 | 0.9[a] | 36.4[ab] |
| with | 0 | 121.5 | 46.8 | 18.0 | 7.6 | 1.3[c] | 1.4 | 1.19 | 45.1 | 38.6 | 14.8[ab] | 6.2 | 1.1[b] | 1.1 | 0.6[b] | 36.9[a] |
| | 0.1 | 118.7 | 45.3 | 17.2 | 7.7 | 2.5[ab] | 2.5 | 1.10 | 42.1 | 38.2 | 14.5[b] | 6.5 | 2.1[a] | 2.1 | 0.9[a] | 35.4[b] |
| SEM[2] | | 1.93 | 0.71 | 0.28 | 0.22 | 0.17 | 0.19 | 0.05 | 0.84 | 0.21 | 0.05 | 0.13 | 0.13 | 0.15 | 0.03 | 0.17 |
| Main effects | | | | | | | | | | | | | | | | |
| *Medication* | | | | | | | | | | | | | | | | |
| without | | 123.9 | 47.9 | 18.3 | 8.2 | 2.1 | 1.4 | 0.99 | 44.8 | 38.6 | 14.8 | 6.6 | 1.7 | 1.1 | 0.7 | 36.1 |
| with | | 120.6 | 46.1 | 17.6 | 7.6 | 1.9 | 1.9 | 1.14 | 43.6 | 38.4 | 14.6 | 6.4 | 1.6 | 1.6 | 0.9 | 36.2 |
| *BGase (%)* | | | | | | | | | | | | | | | | |
| 0 | | 123.4 | 47.5 | 18.2 | 7.9 | 2.0 | 1.4 | 1.19[a] | 45.0 | 38.5 | 14.7 | 6.4 | 1.6 | 1.1 | 0.9 | 36.4 |
| 0.1 | | 120.6 | 46.4 | 17.7 | 7.9 | 2.0 | 1.9 | 0.95[b] | 43.4 | 38.5 | 14.7 | 6.6 | 1.7 | 1.6 | 0.7 | 35.9 |
| *Probability (%)* | | | | | | | | | | | | | | | | |
| Medication | | 0.29 | 0.16 | 0.16 | 0.24 | 0.53 | 0.17 | 0.10 | 0.45 | 0.64 | 0.22 | 0.42 | 0.69 | 0.12 | 0.02 | 0.89 |
| BGase | | 0.43 | 0.41 | 0.39 | 0.99 | 0.98 | 0.17 | 0.01 | 0.30 | 0.94 | 0.79 | 0.54 | 0.77 | 0.13 | 0.01 | 0.16 |
| Medication × BGase | | 0.99 | 0.75 | 0.50 | 0.90 | 0.0003 | 0.10 | 0.09 | 0.39 | 0.36 | 0.01 | 0.84 | 0.002 | 0.06 | 0.04 | 0.001 |

[a-d]Means within a main effect or interaction not sharing a common superscript are significantly different ($P \leq 0.05$).

[1]BGase—β-glucanase; SCFA—short chain fatty acids; Ace—Acetic acid; Pro—Propionic acid; But—Butyric acid; Isob—Isobutyric acid; Val—Valeric acid; Isov—Isovaleric acid; Cap—Caproic acid.

[2]SEM—pooled standard error of mean (n = 12 birds per treatment).

**Table 11. Effects of diet medication and β-glucanase on caecal short chain fatty acids of broiler chickens aged 11 days (Experiment 2).**

| Medication | BGase[1] (%) | SCFA μmol/g of wet caecal content | | | | | | | | Molar percentage of total SCFA | | | | | | |
|---|---|---|---|---|---|---|---|---|---|---|---|---|---|---|---|---|
| | | Total | Ace | Pro | But | Isob | Val | Isov | Cap | Ace | Pro | But | Isob | Val | Isov | Cap |
| without | 0 | 228.6[b] | 134.1[b] | 49.7[b] | 22.7[b] | 7.4[b] | 4.3[b] | 7.4[b] | 2.7[b] | 58.6 | 21.8 | 9.9 | 3.2 | 1.7 | 3.3[a] | 0.1 |
| | 0.1 | 306.6[a] | 176.5[a] | 66.3[a] | 30.0[a] | 9.9[a] | 9.7[a] | 9.8[a] | 4.2[a] | 57.5 | 21.6 | 9.7 | 3.2 | 3.1 | 3.2[ab] | 0.1 |
| with | 0 | 172.8[b] | 100.9[bc] | 36.4[bc] | 17.5[b] | 5.4[c] | 4.6[b] | 5.4[c] | 2.3[b] | 58.3 | 21.1 | 10.1 | 3.1 | 2.7 | 3.1[b] | 0.1 |
| | 0.1 | 171.2[b] | 98.8[c] | 36.7[c] | 16.8[b] | 5.5[c] | 5.4[b] | 5.4[c] | 2.2[b] | 57.7 | 21.4 | 9.8 | 3.2 | 3.1 | 3.2[ab] | 0.1 |
| SEM[2] | | 12.94 | 7.41 | 2.83 | 1.25 | 0.42 | 0.58 | 0.41 | 0.19 | 0.21 | 0.05 | 0.13 | 0.01 | 0.13 | 0.15 | 0.03 |
| Main effects | | | | | | | | | | | | | | | | |
| *Medication* | | | | | | | | | | | | | | | | |
| without | | 267.6 | 155.3 | 58.0 | 26.3 | 8.7 | 7.0 | 8.6 | 3.4 | 58.1 | 21.7[a] | 9.8 | 3.2[a] | 2.4 | 3.2 | 0.1 |
| with | | 172.0 | 99.8 | 36.6 | 17.2 | 5.4 | 5.0 | 5.4 | 2.3 | 58.0 | 21.3[b] | 9.8 | 3.1[b] | 2.9 | 3.1 | 0.1 |
| *BGase (%)* | | | | | | | | | | | | | | | | |
| 0 | | 200.7 | 117.5 | 43.1 | 20.1 | 6.4 | 4.5 | 6.4 | 2.5 | 58.5[a] | 21.5 | 10.0[a] | 3.2 | 2.2[b] | 3.2 | 0.1 |
| 0.1 | | 238.9 | 137.7 | 51.5 | 23.4 | 7.7 | 7.5 | 7.6 | 3.2 | 57.6[b] | 21.5 | 9.8[b] | 3.2 | 3.1[a] | 3.1 | 0.1 |
| *Probability (%)* | | | | | | | | | | | | | | | | |
| Medication | | < .0001 | < .0001 | < .0001 | < .0001 | < .0001 | 0.02 | < .0001 | 0.0002 | 0.68 | 0.01 | 0.09 | 0.01 | 0.14 | 0.01 | 0.57 |
| BGase | | 0.02 | 0.03 | 0.01 | 0.04 | 0.01 | 0.001 | 0.01 | 0.01 | 0.0004 | 0.91 | 0.01 | 0.89 | 0.007 | 0.93 | 0.34 |
| Medication × BGase | | 0.01 | 0.02 | 0.02 | 0.01 | 0.02 | 0.01 | 0.03 | 0.005 | 0.22 | 0.17 | 0.64 | 0.08 | 0.16 | 0.05 | 0.38 |

[a-d]Means within a main effect or interaction not sharing a common superscript are significantly different ($P \leq 0.05$).

[1]BGase—β-glucanase; SCFA—short chain fatty acids; Ace—Acetic acid; Pro—Propionic acid; But—Butyric acid; Isob—Isobutyric acid; Val—Valeric acid; Isov—Isovaleric acid; Cap—Caproic acid.

[2]SEM—pooled standard error of mean (n = 12 birds per treatment).

**Table 12. Effects of diet medication and β-glucanase on ileal short chain fatty acids of broiler chickens aged 33 days (Experiment 2).**

| Medication | BGase[1] (%) | SCFA μmol/g of wet ileal content | | | | | | | | Molar percentage of total SCFA | | | | | | |
|---|---|---|---|---|---|---|---|---|---|---|---|---|---|---|---|---|
| | | Total | Ace | Pro | But | Val | Isov | Cap | Lac | Ace | Pro | But | Val | Isov | Cap | Lac |
| without | 0 | 115.2 | 44.6 | 17.0 | 7.6 | 1.5 | 1.6 | 1.0 | 41.6 | 38.7 | 14.79 | 6.6 | 1.3 | 1.4 | 0.8 | 36.1 |
| | 0.1 | 125.0 | 47.8 | 18.1 | 8.1 | 2.6 | 2.7 | 1.1 | 44.3 | 38.2 | 14.52 | 6.5 | 2.1 | 2.1 | 0.9 | 35.4 |
| with | 0 | 118.9 | 46.0 | 17.5 | 7.8 | 1.7 | 1.9 | 1.0 | 42.7 | 38.7 | 14.74 | 6.6 | 1.4 | 1.6 | 0.8 | 35.9 |
| | 0.1 | 123.0 | 47.1 | 17.9 | 7.5 | 2.6 | 2.6 | 1.1 | 43.8 | 38.3 | 14.60 | 6.1 | 2.1 | 2.1 | 0.9 | 35.6 |
| SEM[2] | | 1.21 | 0.46 | 0.17 | 0.13 | 0.11 | 0.11 | 0.02 | 0.43 | 0.21 | 0.05 | 0.13 | 0.13 | 0.15 | 0.03 | 0.17 |
| Main effects | | | | | | | | | | | | | | | | |
| *Medication* | | | | | | | | | | | | | | | | |
| without | | 120.1 | 46.2 | 17.6 | 7.8 | 2.1 | 2.1 | 1.0 | 42.9 | 38.5 | 14.6 | 6.5 | 1.7 | 1.7 | 0.9 | 35.7 |
| with | | 120.9 | 46.5 | 17.7 | 7.7 | 2.2 | 2.3 | 1.0 | 43.2 | 38.5 | 14.6 | 6.3 | 1.8 | 1.8 | 0.9 | 35.7 |
| *BGase (%)* | | | | | | | | | | | | | | | | |
| 0 | | 117.0[b] | 45.3[b] | 17.2[b] | 7.7 | 1.6[b] | 1.7[b] | 1.0[b] | 42.1[b] | 38.7 | 14.7 | 6.6 | 1.4[b] | 1.5[b] | 0.8 | 36.0[a] |
| 0.1 | | 124.0[a] | 47.5[a] | 18.0[a] | 7.8 | 2.6[a] | 2.6[a] | 1.1[a] | 44.0[a] | 38.3 | 14.5 | 6.3 | 2.1[a] | 2.1[a] | 0.9 | 35.5[b] |
| *Probability* (%) | | | | | | | | | | | | | | | | |
| Medication | | 0.73 | 0.72 | 0.69 | 0.51 | 0.71 | 0.48 | 0.88 | 0.70 | 0.91 | 0.88 | 0.30 | 0.77 | 0.53 | 0.82 | 0.91 |
| BGase | | 0.003 | 0.02 | 0.02 | 0.68 | < .0001 | < .0001 | 0.01 | 0.02 | 0.30 | 0.10 | 0.12 | 0.001 | 0.003 | 0.18 | 0.001 |
| Medication × BGase | | 0.22 | 0.24 | 0.34 | 0.15 | 0.61 | 0.40 | 0.92 | 0.34 | 0.76 | 0.25 | 0.37 | 0.83 | 0.57 | 0.72 | 0.24 |

[a-d]Means within a main effect or interaction not sharing a common superscript are significantly different ($P \leq 0.05$).

[1]BGase—β-glucanase, SCFA—short chain fatty acids; Ace—Acetic acid; Pro—Propionic acid; But—Butyric acid; Isob—Isobutyric acid; Val—Valeric acid; Isov—Isovaleric acid; Cap—Caproic acid.

[2]SEM—pooled standard error of mean (n = 18 birds per treatment).

effect when the enzyme was present. However, the interaction was not clear according to mean separation due to the absence of enzyme effect for proportional isovaleric acid.

Medication and the interactions between medication and BGase did not affect the concentrations and molar percentages of ileal SCFA at d 33 (Table 12). All ileal SCFA concentrations except butyric acid were higher because of BGase use. In addition, the percentages of valeric and isovaleric acids were higher for the 0.1 compared to the 0% BGase treatment. In contrast, the lactic acid percentage was slightly lower with enzyme use.

No effect of the interactions of BGase and medication were found for the concentrations and molar percentages of caecal digesta SCFA at d 33 (Table 13). However, the concentrations of total SCFA and acetic acid were lower in medicated diets. Similarly, all other SCFA levels except butyric acid tended ($P = 0.06$–0.07) to be lower with medication use. The molar percentages of acetic acid decreased, while butyric, valeric ($P = 0.08$) and isovaleric ($P = 0.09$) acids increased with medication use. Enzyme use decreased the molar percentage of acetic acid and increased values for all other SCFA except butyric acid, but minimal changes again, as noted earlier.

Interactions between BGase and medication were not found for the digestive tract pH, except for caecal pH at d 11 (Table 14); pH was lower with the enzyme use, but only in the diets without medication. Medication resulted in higher pH in the crop at d 11 and the ileum at both d 11 and 33. Duodenal and ileal pH was higher with the use of BGase at d 11. Gizzard and caecal pH were lower with the enzyme, and ileal pH was higher with diet BGase at d 33.

**Gastro-intestinal wall histomorphology.** Treatment effects were neither prevalent nor consistent between ages for gastrointestinal wall histomorphology responses (Table 15). At d 11, medication decreased the crypt depth, while β-glucanase decreased villi width. At 33 d,

**Table 13. Effects of diet medication and β-glucanase on caecal short chain fatty acids of broiler chickens aged 33 days (Experiment 2).**

| Medication | BGase[1] (%) | SCFA µmol/g of wet caecal content | | | | | | | | Molar percentage of total SCFA | | | | | | |
|---|---|---|---|---|---|---|---|---|---|---|---|---|---|---|---|---|
| | | Total | Ace | Pro | But | Isob | Val | Isov | Cap | Ace | Pro | But | Isob | Val | Isov | Cap |
| without | 0 | 225.0 | 132.2 | 46.5 | 22.5 | 6.9 | 6.8 | 6.8 | 2.9 | 58.8 | 20.6 | 10.0 | 3.0 | 3.04 | 3.05 | 1.31 |
| | 0.1 | 230.7 | 134.9 | 48.1 | 23.0 | 7.2 | 7.1 | 7.1 | 3.0 | 58.5 | 20.8 | 9.9 | 3.1 | 3.08 | 3.09 | 1.33 |
| with | 0 | 209.8 | 122.6 | 43.5 | 21.4 | 6.5 | 6.4 | 6.4 | 2.7 | 58.4 | 20.7 | 10.2 | 3.1 | 3.07 | 3.07 | 1.32 |
| | 0.1 | 215.5 | 125.3 | 45.1 | 22.0 | 6.7 | 6.6 | 6.7 | 2.8 | 58.1 | 20.9 | 10.2 | 3.1 | 3.10 | 3.11 | 1.33 |
| SEM[2] | | 3.78 | 2.17 | 0.82 | 0.38 | 0.12 | 0.12 | 0.12 | 0.05 | 0.21 | 0.05 | 0.13 | 0.01 | 0.13 | 0.15 | 0.03 |
| Main effects | | | | | | | | | | | | | | | | |
| *Medication* | | | | | | | | | | | | | | | | |
| without | | 227.8[a] | 133.5[a] | 47.3 | 22.7 | 7.0 | 6.9 | 7.0 | 3.0 | 58.6[a] | 20.7 | 10.0[b] | 3.1 | 3.06 | 3.07 | 1.32 |
| with | | 212.6[b] | 124.0[b] | 44.3 | 21.7 | 6.6 | 6.5 | 6.5 | 2.8 | 58.2[b] | 20.8 | 10.2[a] | 3.1 | 3.08 | 3.09 | 1.33 |
| *BGase (%)* | | | | | | | | | | | | | | | | |
| 0 | | 217.4 | 127.4 | 45.0 | 22.0 | 6.7 | 6.6 | 6.6 | 2.8 | 58.6[a] | 20.7[b] | 10.1 | 3.0[b] | 3.05[b] | 3.06[b] | 1.31[b] |
| 0.1 | | 223.1 | 130.1 | 46.6 | 22.5 | 6.9 | 6.8 | 6.9 | 2.9 | 58.3[b] | 20.9[a] | 10.0 | 3.1[a] | 3.09[a] | 3.09[a] | 1.33[a] |
| *Probability* (%) | | | | | | | | | | | | | | | | |
| Medication | | 0.04 | 0.02 | 0.06 | 0.15 | 0.06 | 0.07 | 0.07 | 0.07 | 0.005 | 0.20 | 0.02 | 0.14 | 0.08 | 0.09 | 0.12 |
| BGase | | 0.43 | 0.51 | 0.31 | 0.50 | 0.27 | 0.29 | 0.28 | 0.27 | 0.03 | 0.02 | 0.75 | 0.004 | 0.01 | 0.01 | 0.004 |
| Medication × BGase | | 0.99 | 0.99 | 0.99 | 0.93 | 0.98 | 0.99 | 0.99 | 0.94 | 0.93 | 0.97 | 0.85 | 0.88 | 0.93 | 0.97 | 0.59 |

[a-d]Means within a main effect or interaction not sharing a common superscript are significantly different ($P \leq 0.05$).

[1]BGase—β-glucanase; SCFA—short chain fatty acids; Ace—Acetic acid; Pro—Propionic acid; But—Butyric acid; Isob—Isobutyric acid; Val—Valeric acid; Isov—Isovaleric acid; Cap—Caproic acid.

[2]SEM—pooled standard error of mean (n = 18 birds per treatment).

medication increased the number of acidic and decreased the number of mixed goblet cells per villus. The medication also increased the villi height to crypt depth ratio.

**Digestive tract morphology.** Interactions were found between medication and BGase for the empty proportional weights of the duodenum, jejunum, small intestine and caeca at d 11 (Table 16). However, the interaction for cecal empty weight was not clear based on mean separation since the enzyme effect is absent regardless of medication. For all segments, feeding diets without medication or enzyme resulted in the heaviest weights. Using an enzyme in non-medicated diets reduced the segment weights (jejunum and small intestine), while enzyme use in diets with medication did not affect empty weight. Feeding an enzyme reduced the proventriculus empty weight and medication reduced the ileum weight. The length of the jejunum, ileum, small intestine and caeca were shorter with medication use. The dietary enzyme reduced the length of the jejunum and the small intestine.

Diet medication decreased the empty proportional weights of the duodenum, jejunum, ileum, small intestine and colon, and decreased the lengths of the same digestive tract segments in 33 d old broilers (Table 17). Dietary BGase resulted in lower empty weights for the crop, ileum and small intestine; the enzyme also reduced the lengths of the duodenum and ileum. Interactions between BGase and medication were found for the empty jejunum weight, and the lengths of the jejunum and small intestine. However, mean separation failed to establish the interaction for jejunum weight due to the absence of enzyme effect regardless of medication. For the interactions, enzyme use resulted in smaller tissues (only the jejunum and small intestine lengths) when non-medicated diets were fed but had no effect when diets contained medication. Medication resulted in smaller digestive tract segments in these interactions.

**Table 14. Effects of diet medication and diet on gastro-intestinal pH of broiler chickens (Experiment 2).**

| Medication | [1]BGase (%) | pH | | | | | | | | | | | |
|---|---|---|---|---|---|---|---|---|---|---|---|---|---|
| | | d 11 | | | | | | d 33 | | | | | |
| | | Crop | Gizzard | Duodenum | Jejunum | Ileum | Caeca | Crop | Gizzard | Duodenum | Jejunum | Ileum | Caeca |
| without | 0 | 4.78 | 2.81 | 5.88 | 5.91 | 6.29 | 6.36[a] | 4.94 | 3.67 | 6.15 | 5.93 | 6.50 | 6.22 |
| | 0.1 | 4.62 | 2.41 | 5.99 | 5.92 | 6.61 | 5.78[b] | 4.84 | 3.44 | 6.01 | 5.99 | 6.94 | 6.03 |
| with | 0 | 4.93 | 2.49 | 5.90 | 5.90 | 6.62 | 5.70[b] | 5.01 | 3.75 | 6.18 | 5.97 | 7.20 | 6.19 |
| | 0.1 | 5.09 | 2.55 | 6.06 | 6.01 | 6.97 | 5.77[b] | 4.91 | 3.28 | 6.18 | 5.99 | 7.39 | 5.96 |
| SEM[2] | | 0.052 | 0.057 | 0.024 | 0.018 | 0.053 | 0.061 | 0.052 | 0.057 | 0.024 | 0.018 | 0.053 | 0.061 |
| Main effects | | | | | | | | | | | | | |
| *Medication* | | | | | | | | | | | | | |
| without | | 4.70[b] | 2.61 | 5.94 | 5.92 | 6.45[b] | 6.07 | 4.89 | 3.55 | 6.08 | 5.96 | 6.72[b] | 6.12 |
| with | | 5.01[a] | 2.52 | 5.98 | 5.96 | 6.80[a] | 5.74 | 4.96 | 3.52 | 6.18 | 5.98 | 7.30[a] | 6.08 |
| *BGase (%)* | | | | | | | | | | | | | |
| 0 | | 4.85 | 2.65 | 5.89[b] | 5.91 | 6.45[b] | 6.03 | 4.97 | 3.71[a] | 6.16 | 5.95 | 6.85[b] | 6.21[a] |
| 0.1 | | 4.86 | 2.48 | 6.03[a] | 5.97 | 6.79[a] | 5.78 | 4.87 | 3.36[b] | 6.09 | 5.99 | 7.17[a] | 5.99[b] |
| *Probability* | | | | | | | | | | | | | |
| Medication | | 0.001 | 0.41 | 0.33 | 0.25 | 0.0001 | 0.001 | 0.46 | 0.71 | 0.09 | 0.61 | < .0001 | 0.65 |
| BGase | | 0.97 | 0.12 | 0.004 | 0.10 | 0.0002 | 0.01 | 0.29 | 0.001 | 0.22 | 0.28 | 0.0007 | 0.04 |
| Medication × BGase | | 0.10 | 0.04 | 0.66 | 0.14 | 0.84 | 0.002 | 0.98 | 0.24 | 0.21 | 0.61 | 0.16 | 0.88 |

[a-b]Means within a main effect or interaction not sharing a common superscript are significantly different ($P \leq 0.05$).

[1]BGase—β-glucanase

[2] SEM—pooled standard error of mean (d 11; n = 12 birds per treatment, d 33; n = 18 birds per treatment).

**Measurements of the contents of the digestive tract, and digestive organ morphology.** The content weight of the small intestine was lower, with the addition of BGase to the diets without medication (Table 18). Medication reduced the content weight of the crop and caeca, while BGase reduced the content weight of the gizzard, jejunum, ileum and colon. Diet medication reduced the pancreas weight, and diet enzyme increased liver weight and decreased pancreas weight.

The content weights of the duodenum and colon decreased with the use of BGase at d 33 (Table 19). Medication similarly decreased the content weight of the duodenum. Interactions between medication and enzyme were found for the content weights of the gizzard ($P = 0.06$), jejunum, ileum, small intestine and colon ($P = 0.06$). For the jejunum, ileum, small intestine and colon segments, the enzyme reduced weights in non-medicated diets but did not affect content weights in medication presence. For gizzard content weights, enzyme tended to increase and decrease values in diets without and with medication, respectively. An interaction was also found for liver weight. The largest weight was found for the birds fed diets with no medication or enzyme; the addition of enzyme to the unmedicated diet resulted in lower weight, and the liver weights for medicated diets were smallest and unaffected by the enzyme in the diet.

**Body weight gain, feed intake and feed to gain ratio.** Interactions between BGase and medication were significant for body weight gain for all periods (Table 20), but the nature of the response changed with age. From 0–11 d, medication increased gain in the birds given diets with or without BGase, while enzyme did not affect the gain. Weight gain from 11 to 22 d was increased by enzyme regardless of diet medication, and medication increased the gain in the treatments with or without BGase. From 22–32 d, enzyme increased gain in the non-

**Table 15. Effects of medication and β-glucanase on histomorphology responses in the ileum of broiler chickens (Experiment 2).**

| Medication | BGase[1] (%) | d 11 | | | | | | | d 33 | | | | | | |
|---|---|---|---|---|---|---|---|---|---|---|---|---|---|---|---|
| | | Villi height (μm) | Villi width (μm) | Number of goblet cells/ villus | | | Crypt depth (μm) | Villi height: Crypt depth | Villi height (μm) | Villi width (μm) | Number of goblet cells/ villus | | | Crypt depth (μm) | Villi height: Crypt depth |
| | | | | Acidic | Neutral | Mixed | | | | | Acidic | Neutral | Mixed | | |
| without | 0 | 402 | 101 | 30 | 12 | 4 | 136 | 3.1 | 657 | 117 | 77 | 20 | 7 | 134 | 5 |
| | 0.1 | 446 | 92 | 35 | 17 | 6 | 139 | 3.2 | 656 | 115 | 63 | 20 | 9 | 160 | 4 |
| with | 0 | 405 | 104 | 41 | 11 | 5 | 107 | 3.7 | 734 | 113 | 87 | 20 | 6 | 136 | 5 |
| | 0.1 | 383 | 88 | 37 | 15 | 4 | 121 | 3.2 | 746 | 124 | 91 | 25 | 3 | 143 | 5 |
| SEM[2] | | 22.27 | 2.20 | 2.59 | 1.30 | 0.46 | 5.21 | 0.19 | 23.26 | 2.60 | 4.44 | 1.74 | 0.96 | 4.61 | 0.18 |
| Main effects | | | | | | | | | | | | | | | |
| *Medication* | | | | | | | | | | | | | | | |
| without | | 424 | 97 | 32 | 14 | 5 | 137[a] | 3.1 | 656 | 116 | 70[b] | 20 | 8[a] | 147 | 4[b] |
| with | | 394 | 96 | 39 | 13 | 5 | 114[b] | 3.4 | 740 | 118 | 89[a] | 22 | 4[b] | 140 | 5[a] |
| *BGase (%)* | | | | | | | | | | | | | | | |
| 0 | | 404 | 102[a] | 35 | 11 | 5 | 121 | 3.4 | 695 | 115 | 82 | 20 | 6 | 135 | 5 |
| 0.1 | | 414 | 90[b] | 36 | 16 | 5 | 130 | 3.2 | 701 | 120 | 77 | 22 | 6 | 151 | 4 |
| *Probability* | | | | | | | | | | | | | | | |
| Medication | | 0.54 | 0.91 | 0.21 | 0.56 | 0.82 | 0.01 | 0.41 | 0.07 | 0.62 | 0.03 | 0.48 | 0.04 | 0.39 | 0.03 |
| BGase | | 0.83 | 0.01 | 0.96 | 0.08 | 0.96 | 0.32 | 0.58 | 0.90 | 0.29 | 0.52 | 0.51 | 0.98 | 0.06 | 0.13 |
| Medication × BGase | | 0.50 | 0.43 | 0.39 | 0.94 | 0.22 | 0.53 | 0.43 | 0.88 | 0.17 | 0.28 | 0.59 | 0.21 | 0.25 | 0.17 |

[a-b] Means within a main effect or interaction not sharing a common superscript are significantly different ($P \leq 0.05$).

[1]BGase—β-glucanase.

[2]SEM—pooled standard error of mean (n = 6 birds per treatment).

medicated diets but had no effect when diets contain medication. Overall, weight gain (0–32 d) was increased by enzyme use, regardless of diet medication, but to a greater extent in the absence of medication.

Medication and enzyme use increased feed intake from 0–11 d, and medication similarly increased feed intake from 11–22 d. Interactions between medication and enzyme were significant from 22–32 d and approached significance ($P = 0.06$) for the overall experiment. Medication increased the feed intake in the treatments with BGase from 22–32 d. In the overall period, medication increased the feed consumption.

Interactions were found between medication and BGase for F:G in all periods. Medication improved the feed efficiency throughout the trial, but as was the case for body weight gain, the nature of the interaction with enzyme use changed with bird age. During the 0–11 d period, F: G worsened with enzyme use when birds were fed non-medicated diets but had no effect when the medication was used. For the remainder of the periods, including the total trial, enzyme improved F:G in birds fed non-medicated diets but did not affect broilers consuming medicated diets.

The total mortality of the trial was 3.9% and not affected by HB or BGase. The causes of death include infectious (yolk sac infection, coccidiosis, systemic), metabolic (sudden death syndrome, heart failure), and other diseases in both experiments. The mortality attributed to coccidiosis (by necropsy) was identified as 4.3% of the total mortality. However, 46.7% of the total mortality was detected as a systemic infection, including necrotic enteritis. Subclinical coccidiosis in the birds may damage the intestinal epithelial membrane and enhance systemic infections due to bacterial translocation.

**Table 16. Effects of diet medication and β-glucanase on gastro-intestinal tissue weights and lengths (proportional to body weight) of broiler chickens at day 11 (Experiment 2).**

| Medication | BGase[1] (%) | | Empty weight (%) | | | | | | | | | Length (cm/100g) | | | | | |
|---|---|---|---|---|---|---|---|---|---|---|---|---|---|---|---|---|---|
| | | Crop | Proven | Gizzard | Duodenum | Jejunum | Ileum | SI | Caeca | Colon | | Duodenum | Jejunum | Ileum | SI | Caeca | Colon |
| without | 0 | 0.53 | 0.83 | 2.63 | 1.92ᵃ | 2.97ᵃ | 2.11 | 7.00ᵃ | 0.66ᵃ | 0.26 | | 7.22 | 17.42 | 15.66 | 40.29 | 5.40 | 1.39 |
| | 0.1 | 0.48 | 0.79 | 2.61 | 1.77ᵃᵇ | 2.63ᵇ | 1.88 | 6.27ᵇ | 0.60ᵃᵇ | 0.22 | | 6.90 | 15.16 | 14.97 | 37.02 | 5.24 | 1.36 |
| with | 0 | 0.46 | 0.87 | 2.69 | 1.51ᵇ | 2.40ᵇ | 1.74 | 5.65ᶜ | 0.50ᵇ | 0.25 | | 7.13 | 15.45 | 13.56 | 36.14 | 4.62 | 1.40 |
| | 0.1 | 0.48 | 0.77 | 2.54 | 1.69ᵃᵇ | 2.67ᵇ | 1.78 | 6.13ᵇᶜ | 0.62ᵃᵇ | 0.25 | | 6.11 | 14.64 | 13.75 | 34.49 | 4.99 | 1.34 |
| SEM[2] | | 0.018 | 0.018 | 0.043 | 0.039 | 0.053 | 0.043 | 0.109 | 0.020 | 0.006 | | 0.219 | 0.273 | 0.329 | 0.584 | 0.121 | 0.035 |
| Main effects | | | | | | | | | | | | | | | | | |
| *Medication* | | | | | | | | | | | | | | | | | |
| without | | 0.50 | 0.81 | 2.62 | 1.84 | 2.80 | 2.00ᵃ | 6.64 | 0.63 | 0.24 | | 7.06 | 16.29ᵃ | 15.31ᵃ | 38.65ᵃ | 5.32ᵃ | 1.37 |
| with | | 0.47 | 0.82 | 2.62 | 1.60 | 2.54 | 1.76ᵇ | 5.89 | 0.56 | 0.25 | | 6.62 | 15.05ᵇ | 13.65ᵇ | 35.31ᵇ | 4.80ᵇ | 1.37 |
| *BGase (%)* | | | | | | | | | | | | | | | | | |
| 0 | | 0.49 | 0.85ᵃ | 2.66 | 1.72 | 2.69 | 1.93 | 6.33 | 0.58 | 0.25 | | 7.17 | 16.43ᵃ | 14.61 | 38.21ᵃ | 5.01 | 1.39 |
| 0.1 | | 0.48 | 0.78ᵇ | 2.58 | 1.73 | 2.65 | 1.83 | 6.20 | 0.61 | 0.24 | | 6.50 | 14.90ᵇ | 14.36 | 35.76ᵇ | 5.12 | 1.35 |
| *Probability* | | | | | | | | | | | | | | | | | |
| Medication | | 0.16 | 0.77 | 0.92 | 0.0009 | 0.001 | 0.003 | < .0001 | 0.07 | 0.44 | | 0.26 | 0.004 | 0.01 | 0.001 | 0.03 | 0.92 |
| BGase | | 0.70 | 0.04 | 0.29 | 0.90 | 0.62 | 0.19 | 0.42 | 0.43 | 0.11 | | 0.09 | 0.0007 | 0.69 | 0.01 | 0.65 | 0.41 |
| Medication × BGase | | 0.15 | 0.42 | 0.41 | 0.02 | 0.0004 | 0.08 | 0.0005 | 0.01 | 0.15 | | 0.36 | 0.08 | 0.48 | 0.40 | 0.26 | 0.74 |

ᵃ⁻ᵇMeans within a main effect or interaction not sharing a common superscript are significantly different ($P \leq 0.05$).

[1]BGase—β-glucanase; Proven—proventriculus; SI—small intestine.

[2]SEM—pooled standard error of mean (n = 12 birds per treatment).

## Discussion

With minor exceptions, all three molecular weight responses for soluble ileal digesta β-glucan were lower with the enzyme use, which confirms exogenous BGase mediates the depolymerization of HB β-glucan in broiler chickens. In addition, the reduction of MW-10% with BGase in both experiments further supports β-glucan depolymerization since it demonstrates the increased proportion of small molecular weight soluble β-glucan in ileal digesta. Overall, the response for Mp was similar in both experiments, which indicates that β-glucan depolymerization is independent of the vaccination status of the animal. Further, Mw from Experiment 1 also supports the reduction of molecular weight in the ileal digesta soluble β-glucan with the use of BGase. In contrast, BGase increased Mw at d 33 (numerically increased at d 11) in Experiment 2 in the treatments with antibiotics. The reason for the increased β-glucan Mw is unknown but could relate to the aggregation of smaller weight β-glucan molecules [33] [34, 35] or enzyme-mediated release of higher molecular weight, insoluble β-glucan that had not yet been depolymerized. The release of higher molecular weight β-glucan would be the more credible explanation since the increased Mw has been only observed in Experiment 2, which might be affected by the bird age. The reduction of β-glucan molecular weight and the increased proportion of small molecular weight soluble β-glucan encourage the assessment of performance and digestive tract characteristics due to the potentially increased fermentation of small molecular weight β-glucan. Further, the proportion of small molecular weight β-glucan is an important assessment since chicken microbiota preferred small molecular sugars and peptides over complex polysaccharides and proteins in a study that investigated the utilization of nutrients by chicken caecal and human faecal microbes using an *in vitro* assay [36].

**Table 17. Effects of diet medication and β-glucanase on gastro-intestinal tissue weights and lengths (proportional to body weight) of broiler chickens at day 33 (Experiment 2).**

| Medication | BGase[1] (%) | Empty weight (%) | | | | | | | | | Length (cm/100g) | | | | | |
|---|---|---|---|---|---|---|---|---|---|---|---|---|---|---|---|---|
| | | Crop | Proven | Gizzard | Duo | Jejunum | Ileum | SI | Caeca | Colon | Duo | Jejunum | Ileum | SI | Caeca | Colon |
| without | 0 | 0.30 | 0.38 | 1.12 | 0.87 | 1.64[a] | 1.13 | 3.64 | 0.37 | 0.17 | 1.80 | 4.49[a] | 4.42 | 10.70[a] | 0.63 | 0.41 |
| | 0.1 | 0.29 | 0.39 | 1.23 | 0.86 | 1.53[a] | 1.00 | 3.38 | 0.38 | 0.15 | 1.63 | 3.88[b] | 3.86 | 9.37[b] | 0.71 | 0.38 |
| with | 0 | 0.33 | 0.44 | 1.14 | 0.71 | 1.24[b] | 0.98 | 2.92 | 0.35 | 0.15 | 1.57 | 3.43[c] | 3.36 | 8.35[c] | 0.60 | 0.32 |
| | 0.1 | 0.27 | 0.36 | 1.16 | 0.70 | 1.28[b] | 0.92 | 2.90 | 0.37 | 0.15 | 1.47 | 3.40[c] | 3.34 | 8.20[c] | 0.64 | 0.35 |
| SEM[2] | | 0.006 | 0.015 | 0.022 | 0.014 | 0.029 | 0.018 | 0.051 | 0.008 | 0.004 | 0.029 | 0.078 | 0.089 | 0.172 | 0.031 | 0.010 |
| Main effects | | | | | | | | | | | | | | | | |
| *Medication* | | | | | | | | | | | | | | | | |
| without | | 0.29 | 0.38 | 1.17 | 0.86[a] | 1.58 | 1.06[a] | 3.51[a] | 0.38 | 0.16[a] | 1.71[a] | 4.19 | 4.14[a] | 10.03 | 0.67 | 0.40[a] |
| with | | 0.30 | 0.40 | 1.15 | 0.70[b] | 1.26 | 0.95[b] | 2.91[b] | 0.36 | 0.15[b] | 1.52[b] | 3.41 | 3.35[b] | 8.27 | 0.62 | 0.33[b] |
| *BGase (%)* | | | | | | | | | | | | | | | | |
| 0 | | 0.31[a] | 0.41 | 1.13 | 0.79 | 1.44 | 1.05[a] | 3.28[a] | 0.36 | 0.16 | 1.68[a] | 3.96 | 3.89[a] | 9.52 | 0.62 | 0.37 |
| 0.1 | | 0.28[b] | 0.38 | 1.20 | 0.78 | 1.40 | 0.96[b] | 3.14[b] | 0.38 | 0.15 | 1.55[b] | 3.64 | 3.60[b] | 8.78 | 0.67 | 0.36 |
| *Probability* | | | | | | | | | | | | | | | | |
| Medication | | 0.80 | 0.57 | 0.62 | < .0001 | < .0001 | 0.0005 | < .0001 | 0.36 | 0.01 | 0.0003 | < .0001 | < .0001 | < .0001 | 0.15 | 0.0004 |
| BGase | | 0.005 | 0.27 | 0.12 | 0.55 | 0.33 | 0.005 | 0.04 | 0.22 | 0.11 | 0.01 | 0.009 | 0.04 | 0.004 | 0.11 | 0.88 |
| Medication × BGase | | 0.12 | 0.20 | 0.31 | 0.83 | 0.05 | 0.28 | 0.10 | 0.88 | 0.15 | 0.47 | 0.01 | 0.06 | 0.02 | 0.68 | 0.09 |

[a-c]Means within a main effect or interaction not sharing a common superscript are significantly different ($P \leq 0.05$).

[1]BGase—β-glucanase; Proven—proventriculus; Duo—duodenum; SI—small intestine.

[2]SEM—pooled standard error of mean (n = 18 birds per treatment).

**Table 18. Effects of diet medication and β-glucanase on gastro-intestinal content and organ weights as a percentage of body weight of broiler chickens at day 11 (Experiment 2).**

| Medication | BGase[1] (%) | Content | | | | | | | | | Weight | | |
|---|---|---|---|---|---|---|---|---|---|---|---|---|---|
| | | Crop | Proventriculus | Gizzard | Duodenum | Jejunum | Ileum | SI | Caeca | Colon | Liver | Spleen | Pancreas |
| without | 0 | 0.48 | 0.06 | 0.89 | 0.08 | 0.59 | 0.60 | 1.26[a] | 0.08 | 0.06 | 4.05 | 0.13 | 0.57 |
| | 0.1 | 0.54 | 0.05 | 0.81 | 0.05 | 0.45 | 0.41 | 0.89[c] | 0.11 | 0.04 | 4.74 | 0.11 | 0.50 |
| with | 0 | 0.29 | 0.11 | 0.99 | 0.05 | 0.53 | 0.51 | 1.08[b] | 0.07 | 0.07 | 4.19 | 0.13 | 0.50 |
| | 0.1 | 0.37 | 0.06 | 0.73 | 0.04 | 0.45 | 0.44 | 0.93[bc] | 0.07 | 0.05 | 4.48 | 0.12 | 0.49 |
| SEM[2] | | 0.035 | 0.008 | 0.034 | 0.006 | 0.018 | 0.018 | 0.727 | 0.006 | 0.004 | 0.070 | 0.004 | 0.011 |
| Main effects | | | | | | | | | | | | | |
| *Medication* | | | | | | | | | | | | | |
| without | | 0.51[a] | 0.05 | 0.85 | 0.06 | 0.52 | 0.50 | 1.08 | 0.09[a] | 0.05 | 4.39 | 0.12 | 0.53[a] |
| with | | 0.33[b] | 0.08 | 0.86 | 0.04 | 0.49 | 0.47 | 1.00 | 0.07[b] | 0.06 | 4.34 | 0.12 | 0.50[b] |
| *BGase (%)* | | | | | | | | | | | | | |
| 0 | | 0.38 | 0.08 | 0.94[a] | 0.06 | 0.56[a] | 0.55[a] | 1.17 | 0.07 | 0.07[a] | 4.12[b] | 0.13 | 0.54[a] |
| 0.1 | | 0.46 | 0.05 | 0.77[b] | 0.04 | 0.45[b] | 0.42[b] | 0.91 | 0.09 | 0.04[b] | 4.61[a] | 0.11 | 0.50[b] |
| *Probability* | | | | | | | | | | | | | |
| Medication | | 0.008 | 0.08 | 0.89 | 0.09 | 0.29 | 0.36 | 0.11 | 0.03 | 0.09 | 0.63 | 0.64 | 0.04 |
| BGase | | 0.26 | 0.08 | 0.009 | 0.06 | < .0001 | 0.0001 | < .0001 | 0.20 | 0.005 | 0.0002 | 0.10 | 0.03 |
| Medication × BGase | | 0.85 | 0.15 | 0.15 | 0.22 | 0.16 | 0.06 | 0.02 | 0.22 | 0.91 | 0.09 | 0.57 | 0.13 |

[a-b]Means within a main effect or interaction not sharing a common superscript are significantly different ($P \leq 0.05$).

[1]BGase—β-glucanase; SI—small intestine

[2]SEM—pooled standard error of mean (n = 12 birds per treatment).

**Table 19. Effects of diet medication and β-glucanase on gastro-intestinal content and organ weights as a percentage of body weight of broiler chickens at day 33 (Experiment 2).**

| Medication | BGase[1] (%) | Content | | | | | | | | | Weight | | |
|---|---|---|---|---|---|---|---|---|---|---|---|---|---|
| | | Crop | Proventriculus | Gizzard | Duodenum | Jejunum | Ileum | SI | Caeca | Colon | Liver | Spleen | Pancreas |
| without | 0 | 1.54 | 0.11 | 1.18 | 0.12 | 1.31[a] | 1.49[a] | 2.91[a] | 0.27 | 0.23 | 3.16[a] | 0.12 | 0.27 |
| | 0.1 | 1.44 | 0.06 | 1.33 | 0.09 | 0.86[b] | 0.97[b] | 1.91[b] | 0.32 | 0.14 | 2.88[b] | 0.12 | 0.27 |
| with | 0 | 1.46 | 0.34 | 1.56 | 0.08 | 1.03[b] | 1.12[b] | 2.21[b] | 0.25 | 0.17 | 2.57[c] | 0.12 | 0.26 |
| | 0.1 | 1.11 | 0.07 | 1.24 | 0.07 | 0.95[b] | 0.91[b] | 1.92[b] | 0.27 | 0.17 | 2.58[c] | 0.12 | 0.26 |
| SEM[2] | | 0.096 | 0.043 | 0.060 | 0.006 | 0.039 | 0.050 | 0.084 | 0.015 | 0.011 | 0.040 | 0.004 | 0.005 |
| Main effects | | | | | | | | | | | | | |
| *Medication* | | | | | | | | | | | | | |
| without | | 1.49 | 0.09 | 1.26 | 0.10[a] | 1.08 | 1.23 | 2.41 | 0.29 | 0.18 | 3.02 | 0.12 | 0.27 |
| with | | 1.28 | 0.20 | 1.40 | 0.07[b] | 0.99 | 1.02 | 2.07 | 0.26 | 0.17 | 2.57 | 0.12 | 0.26 |
| *BGase (%)* | | | | | | | | | | | | | |
| 0 | | 1.50 | 0.23 | 1.37 | 0.10[a] | 1.17 | 1.31 | 2.56 | 0.26 | 0.20[a] | 2.86 | 0.12 | 0.26 |
| 0.1 | | 1.27 | 0.06 | 1.29 | 0.08[b] | 0.90 | 0.94 | 1.91 | 0.29 | 0.15[b] | 2.73 | 0.12 | 0.26 |
| *Probability* | | | | | | | | | | | | | |
| Medication | | 0.28 | 0.16 | 0.22 | 0.006 | 0.15 | 0.009 | 0.01 | 0.21 | 0.61 | <.0001 | 0.54 | 0.13 |
| BGase | | 0.24 | 0.06 | 0.46 | 0.02 | 0.0002 | <.0001 | <.0001 | 0.20 | 0.03 | 0.01 | 0.93 | 0.81 |
| Medication × BGase | | 0.52 | 0.19 | 0.06 | 0.37 | 0.006 | 0.04 | 0.007 | 0.52 | 0.06 | 0.01 | 0.93 | 0.90 |

[a-c]Means within a main effect or interaction not sharing a common superscript are significantly different ($P \leq 0.05$).

[1]BGase—β-glucanase; SI—small intestine

[2]SEM—pooled standard error of mean (n = 18 birds per treatment).

**Table 20. Effects of diet medication and β-glucanase on body weight gain, feed intake and feed efficiency of broiler chickens vaccinated for coccidiosis (Experiment 2).**

| Medication | BGase[1] (%) | BWG (kg) | | | | FI (kg) | | | | F:G | | | |
|---|---|---|---|---|---|---|---|---|---|---|---|---|---|
| | | d 0–11 | d 11–22 | d 22–32 | d 0–32 | d 0–11 | d 11–22 | d 22–32 | d 0–32 | d 0–11 | d 11–22 | d 22–32 | d 0–32 |
| without | 0 | 0.243[b] | 0.562[d] | 0.788[c] | 1.594[d] | 0.328 | 0.979 | 1.540[bc] | 2.846 | 1.321[b] | 1.617[a] | 1.939[a] | 1.721[a] |
| | 0.1 | 0.236[b] | 0.622[c] | 0.881[b] | 1.740[c] | 0.331 | 0.982 | 1.499[c] | 2.813 | 1.372[a] | 1.471[b] | 1.688[b] | 1.561[b] |
| with | 0 | 0.262[a] | 0.675[b] | 0.963[a] | 1.900[b] | 0.331 | 1.049 | 1.581[ab] | 2.961 | 1.242[c] | 1.429[b] | 1.627[c] | 1.497[c] |
| | 0.1 | 0.270[a] | 0.702[a] | 0.981[a] | 1.954[a] | 0.339 | 1.071 | 1.588[a] | 2.998 | 1.236[c] | 1.423[b] | 1.593[c] | 1.479[c] |
| SEM[2] | | 0.002 | 0.640 | 0.904 | 0.025 | 0.002 | 0.008 | 0.009 | 0.017 | 0.011 | 0.015 | 0.024 | 0.017 |
| Main effects | | | | | | | | | | | | | |
| *Medication* | | | | | | | | | | | | | |
| without | | 0.240 | 0.591 | 0.835 | 1.667 | 0.329[b] | 0.981[b] | 1.520 | 2.829[b] | 1.347 | 1.544 | 1.813 | 1.641 |
| with | | 0.266 | 0.689 | 0.972 | 1.927 | 0.335[a] | 1.060[a] | 1.584 | 2.905[a] | 1.239 | 1.426 | 1.610 | 1.488 |
| *BGase (%)* | | | | | | | | | | | | | |
| 0 | | 0.252 | 0.618 | 0.876 | 1.747 | 0.329[b] | 1.014 | 1.560 | 2.904 | 1.282 | 1.523 | 1.783 | 1.609 |
| 0.1 | | 0.253 | 0.662 | 0.931 | 1.847 | 0.335[a] | 1.027 | 1.544 | 2.905 | 1.304 | 1.447 | 1.641 | 1.520 |
| *Probability* | | | | | | | | | | | | | |
| Medication | | <.0001 | <.0001 | <.0001 | <.0001 | 0.01 | <.0001 | <.0001 | <.0001 | <.0001 | <.0001 | <.0001 | <.0001 |
| BGase | | 0.77 | <.0001 | <.0001 | <.0001 | 0.01 | 0.18 | 0.14 | 0.92 | 0.01 | <.0001 | <.0001 | <.0001 |
| Medication × BGase | | 0.006 | 0.02 | 0.001 | 0.002 | 0.29 | 0.33 | 0.04 | 0.06 | 0.001 | <.0001 | <.0001 | <.0001 |

[a-c]Means within a main effect or interaction not sharing a common superscript are significantly different ($P \leq 0.05$).

[1]BGase—β-glucanase; BWG—body weight gain; FI—feed intake; F:G—feed to gain ratio.

[2]SEM—pooled standard error of mean (n = 9 pens per treatment).

The molecular weight values were numerically lower at d 33 compared to d 11 in Experiment 2, which might be associated with an age-related adaptation of gut microbiota to utilize fibre [37]. Further, molecular weight responses were lower in Experiment 1 compared to both ages in Experiment 2. Although the experiments cannot be compared statistically, it does draw attention to experimental variation. The analyses of samples were completed at three different times. However, the probability that analytical error accounted for the variation is unlikely because the determination of β-glucan molecular weight using size exclusion chromatography and Calcofluor post-column derivatization is a well-established technique in food science [29], and all laboratory work was completed in the same lab by the senior author. A more plausible explanation for the difference relates to variation in β-glucan characteristics in the barley samples. The birds were fed diets containing CDC Fibar in both experiments; however, the samples were different in the two experiments. Although they were the same cultivar, environmental conditions such as germination may have impacted β-glucan molecular weight. High moisture content in the environment might activate endogenous enzymes in barley and degrade non-starch polysaccharides, including β-glucan, supported by the improved nutritive value of barley with water treatment [38]. Moreover, the molecular weight differences in the two experiments could be attributed to the resident gut microbiota being markedly different between the studies that could harbor different β-glucanase capabilities. The variable gut microbiota composition among the broiler chickens derived from the same breeder flock and raised under the same conditions, including diets, support the difference in microbial enzyme activity [39]. The BGase effect on the reduction of ileal β-glucan molecular weight in this study is in agreement with previous results from our lab [40].

The molecular weight responses in the two experiments decreased with medication when there was no added BGase in the diet, which is an unexpected finding since the medication does not contain endo-β-glucanase activity. It is possibly due to the effect of the antibiotics on modification of the gastro-intestinal microbial population [41–43], resulting in microbiota with an increased capacity to degrade β-glucan into low molecular weight polysaccharides and oligosaccharides. *In vitro* studies have demonstrated that strict anaerobic caecal microbiota, including *Bacteroides ovatus*, *B. uniformis*, *B. capillosus*, *Enterococcus faecium*, *Clostridium perfringens* and *Streptococcus* strains in broiler chickens are capable of degrading mixed-linked β-glucan [44]. However, medication was not able to breakdown high molecular weight β-glucan to the same extent as BGase. Exogenous BGase depolymerizes high molecular weight soluble β-glucan into low molecular weight β-glucan in the ileal digesta, which leads to a reduction of viscosity of the ileal supernatant in broiler chickens. However, the medication did not affect viscosity of the ileal supernatant in broiler chickens, although the molecular weight was reduced with the addition of antibiotics to the broiler diets.

Overall, BGase appears to reduce the empty weights, lengths, and content weights in the digestive tract segments, which agrees with previous broiler research that used the same diets but without medication [45]. The size reduction coincides with increased digestive efficiency associated with enzyme use reported previously [46, 47]. Medication decreased the empty weights and lengths from the duodenum to colon and the digestive tract segments' content weights. The reduction of digestive tract size and content follows previous research that used in-feed antibiotics (Bacitracin methylene disalicylate and virginiamycin) in broiler chickens [48]. The use of specific antibiotics in feed reduces the growth of pathogenic bacteria in the digestive tract of chickens through the modification of microbial diversity and relative abundance, and immune status [19, 20], thereby increasing nutrient digestibility. The reduction of relative abundance of gut microbiota reduces the competition with the host and enables the host to extract all the required nutrients, and thereby the digestive tract size might be reduced [49, 50]. Further, diet medication might increase nutrient digestion due to increased utilization

of non-starch polysaccharides by the gut microbiota by selecting a more effective fibre degrading microbiome, supported by β-glucan molecular weight reduction with antibiotics addition to the diets in the current research. The effects of medication on relative digestive tract size and content weights were mostly significant when the HB based diets did not contain BGase since the enzyme also decreased digestive tract size.

Levels of SCFA and pH in the digestive tract were used to estimate the effects of diet BGase and antibiotics on carbohydrate fermentation because diet BGase and medication depolymerized soluble β-glucan in the ileal digesta of broiler chickens. Ileal pH was higher with BGase use at both ages of broiler chickens in Experiment 2. A BGase mediated increase in ileal pH is contradictory to the current hypothesis of an enzyme-dependent enhancement of carbohydrate fermentation that might be expected based on a large quantity of low molecular weight β-glucan resulting from β-glucan depolymerization due to enzyme use. The increased ileal pH might relate to the increased feed passage rate from the ileum to caeca with the reduction of soluble β-glucan molecular weight, which permits less time for the bacterial fermentation in the ileum [51]. However, ileal pH is contradictory to total and individual SCFA concentrations in the ileum since BGase increased SCFA levels at d 33 in the current study. A reduction of caecal pH with the enzyme (d 11 without medication; d 33) might indicate increased carbohydrate fermentation in the caeca, which is in agreement with previous research [51]. Further, BGase increased SCFA concentrations in the caeca (d 11 without medication) in the current study, which corresponds with the caecal pH at d 11. Overall, the results suggest BGase has shifted bacterial fermentation from the ileum to caeca in broiler chickens.

The antibiotic-induced modification of the gastro-intestinal microbial population might affect the production of SCFA, which influences the enzyme response on carbohydrate fermentation in broiler chickens. Medication affected intestinal pH in a similar fashion to BGase, and similar to the findings of [52], who found increased ileal pH and lowered caecal pH with the addition of salinomycin and Zn bacitracin to broiler diets. However, diet medication did not affect the concentrations of SCFA in the ileum, whereas it decreased total and most of the individual SCFA concentrations in the caeca in the current study, which is again contradictory to the caecal pH. The reduction of caecal pH might be because of antibiotics reducing protein putrefaction to a greater extent than it did SCFA production in the caeca. However, the concentrations of alkalizing metabolites, including the biogenic amines, are not available in the current study. Nevertheless, the reduction of caecal SCFA concentration was according to the study completed by [52] that used salinomycin in broiler feed. Antibiotics modulate the microbial population of the chicken digestive tract [53, 54], and these microbes might not effectively utilize the fermentable fibre, including β-glucan in the chicken digestive tract due to the lower production of microbial-derived non-starch polysaccharidases. However, it is contradictory to the ileal β-glucan molecular weight findings since medication reduced the molecular weight, demonstrating gastro-intestinal bacteria that could secrete non-starch polysaccharidases. The resulting SCFA might have been immediately utilized by gut microbes to produce other metabolic products and affects the measured levels of SCFA. Of note, the crop pH was higher with diet medication. The crop is colonized by BGase-secreting microbiota [55], and medication modifies the crop microbiota, thereby affects carbohydrate fermentation [56].

Medication increased villus height to crypt depth ratio in the ileum, which indicates increased nutrient absorption surface [57] that eventually leads to the enhancement of nutrient digestion and weight gain of chickens. The effect of diet medication on reducing digestive tract size and content also supports the increased nutrient digestibility, which is indicated by the higher villus height to crypt depth ratio. In addition, medication decreased crypt depth in the ileum. Increased crypt depth indicates high cell proliferation in the intestinal epithelial cells [58], which indicates inflammation in the intestinal mucosa. Thus the mucosa enhances

healing from the inflammatory damage by increasing cell proliferation [59, 60]. Inflammation is a protective mechanism, although uncontrolled and chronic inflammation may damage the affected tissues [61, 62]. Therefore, the reduction of crypt depth is considered as a positive entity that enhances bird health.

Treatment affected SCFA concentrations and intestinal pH in coccidiosis vaccinated broiler chickens, but not in battery-cage raised and unvaccinated birds. Further, the treatment effects were larger for broilers at 11 d (mostly infected with *Eimeria* spp) compared to the same birds at 33 d (mostly recovered from the disease) in the coccidiosis vaccinated study. *Eimeria* spp disturbs the lower gastro-intestinal microbial population in broilers [63, 64] due to the epithelial damage of the intestinal mucosa, which affects SCFA production [65]. On the other hand, a precise estimate of SCFA production might not be measured in the current study due to the digesta collection procedure's limitations. Partial absorption of SCFA to the portal circulation before sample collection leads to under-estimation of the values, and ileal and caecal evacuation that is affected by the time of the sample collection results in individual bird variability in results. In addition, protein fermentation affects digesta pH since some protein fermentation products, including ammonia, indoles, phenols and biogenic amines, increase pH in the digestive tract of chickens [18].

Body weight gain, feed intake and feed efficiency were within the normal range, according to Ross 308 Broiler Performance Objectives [26]. The interaction between BGase and medication was significant for body weight gain and feed efficiency at all the broiler ages in Experiment 2. Over the entire experiment, medication increased both body weight gain and feed efficiency of broilers. However, the medication response was higher without BGase since exogenous BGase increased body weight gain and feed conversion in the current study. Both Zn Bacitracin and ionophore anticoccidials have been classified as growth-promoting drugs in broiler chickens due to their positive impact on body weight gain and feed efficiency [18, 66, 67] because the antibiotics in the diets shift the gastro-intestinal microbial population towards a diversified and potentially beneficial microbiota [19, 68]. Villi height to crypt depth ratio in the ileum increased with medication in the current study, supporting the antibiotics-mediated enhancement of the ileal absorptive surface area. However, total and individual SCFA concentrations in the caeca decreased with the addition of antibiotics, which is contradictory to carbohydrate fermentation induced improvement of physiological and growth responses in the current research.

Beta-glucanase decreased the body weight gain and feed efficiency in the birds aged < 11 d but increased these responses after d 11. These results agree with previous research that used the same diets without medication [45]. The poor weight gain and feed efficiency of younger birds may be attributed to an undesirable effect of the increased quantity of low molecular weight carbohydrates on the gut microbiota due to the coccidiosis vaccination and the immature status of the digestive system and gut microbiota. In the study of [45], BGase dosage of 0.01% increased broiler weight gain and feed efficiency for the same age period (0 to 11 days) compared to 0% BGase. However, 0.1% BGase did not affect the body weight gain and reduced the feed efficiency in the birds aged < 11 d but increased these responses after d 11. Moreover, BGase decreased the total requirement of medication in HB-based diets to achieve a high body weight gain and feed conversion, as the medication response on weight gain and feed efficiency decreased with the addition of BGase to the diets. It demonstrates the ability of BGase to partially replace diet medication in HB-based diets to feed broiler chickens. In contrast to the results of Experiment 2, the effects of medication and BGase on body weight gain, feed intake and feed efficiency were not significant in the broiler ages except the period of d 0–7 in Experiment 1, where birds were grown in battery cages without coccidiosis vaccination. The environment of battery cages is relatively hygienic compared to litter floor pens and is

generally considered to present less pathogenic bacterial exposure with the birds. It might be the reason for the fewer effects of medication and enzyme on body weight gain, feed intake and feed efficiency in the battery cage study.

In conclusion, feed BGase and medication can depolymerize high molecular weight soluble β-glucan of HB into low molecular weight β-glucan in the digestive tract of broilers in both experiments; however, the response was higher with BGase compared to medication. The effects of diet medication and BGase on carbohydrate fermentation were not consistent across sample collections in the two experiments according to SCFA levels and intestinal pH, although treatment effects were observed in certain instances. Exogenous BGase and medication increased the growth performance of broiler chickens. Moreover, BGase reduced the necessity of antibiotics and anticoccidials in HB-based diets to achieve a high level of body weight gain and feed efficiency of broiler chickens vaccinated for coccidiosis.

## Supporting information

**S1 Data.**
(XLSX)

## Acknowledgments

The authors would like to acknowledge the Poultry Centre staff at the University of Saskatchewan, and Dawn Abbott and Tracy Exley for their technical support.

## Author Contributions

**Conceptualization:** Namalika D. Karunaratne, Rex W. Newkirk, Henry L. Classen.

**Data curation:** Namalika D. Karunaratne.

**Formal analysis:** Namalika D. Karunaratne.

**Funding acquisition:** Henry L. Classen.

**Investigation:** Namalika D. Karunaratne, Rex W. Newkirk, Henry L. Classen.

**Methodology:** Namalika D. Karunaratne, Rex W. Newkirk, Nancy P. Ames, Andrew G. Van Kessel, Michael R. Bedford, Henry L. Classen.

**Project administration:** Rex W. Newkirk, Henry L. Classen.

**Resources:** Namalika D. Karunaratne, Rex W. Newkirk, Nancy P. Ames, Michael R. Bedford, Henry L. Classen.

**Supervision:** Rex W. Newkirk, Henry L. Classen.

**Visualization:** Namalika D. Karunaratne, Rex W. Newkirk, Henry L. Classen.

**Writing – original draft:** Namalika D. Karunaratne.

**Writing – review & editing:** Namalika D. Karunaratne, Rex W. Newkirk, Nancy P. Ames, Andrew G. Van Kessel, Michael R. Bedford, Henry L. Classen.

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
