## [Decision Letter · Decision Letter 0]

2 Nov 2020

PONE-D-20-20511

Effects of exogenous β-glucanase on ileal digesta soluble β-glucan molecular weight, digestive tract characteristics, and performance of coccidiosis challenged broiler chickens fed hulless barley-based diets with and without medication

PLOS ONE

Dear Dr. Newkirk,

Thank you for submitting your manuscript to PLOS ONE. After careful consideration, we feel that it has merit but does not fully meet PLOS ONE’s publication criteria as it currently stands. Therefore, we invite you to submit a revised version of the manuscript that addresses the points raised during the review process.

There are some issues that should be considered and the entire manuscript needs to be revised. Interpretations about interactions should be reviewed, as there are some with errors. The introduction and discussion be summarized and used a more objective language to make the article more attractive to the reader.

We look forward to receiving your revised manuscript.

Kind regards,

Arda Yildirim, Ph.D.

Academic Editor

PLOS ONE

Additional Editor Comments:

For your guidance, you can check the reviewers' comments. Thank you for giving us the opportunity to consider your work.

Journal Requirements:

"I have read the journal's policy and the authors of this manuscript have the following competing interests: This study was funded in part by Aviagen North America, Sofina Foods Inc., Prairie Pride Natural Foods Ltd., Chicken Farmers of Saskatchewan, Canadian Poultry Research Council, Poultry Industry Council (Canada), Saskatchewan Broiler Hatching Egg Producer’s Marketing Board, Saskatchewan Egg Producers and Saskatchewan Turkey Producers’ Marketing

Board. There are no patents, products in

development or marketed products to declare. This does not alter our adherence to all the PLOS ONE policies on sharing data and materials."

We note that one or more of the authors are employed by a commercial company: AB Vista.

2.1. Please provide an amended Funding Statement declaring this commercial affiliation, as well as a statement regarding the Role of Funders in your study. If the funding organization did not play a role in the study design, data collection and analysis, decision to publish, or preparation of the manuscript and only provided financial support in the form of authors' salaries and/or research materials, please review your statements relating to the author contributions, and ensure you have specifically and accurately indicated the role(s) that these authors had in your study. You can update author roles in the Author Contributions section of the online submission form.

2.2. Please also provide an updated Competing Interests Statement declaring this commercial affiliation along with any other relevant declarations relating to employment, consultancy, patents, products in development, or marketed products, etc.  

3. Please include a copy of Table 10 which you refer to in your text on page 20.

4. We note you have included a table to which you do not refer in the text of your manuscript. Please ensure that you refer to Table 2 in your text; if accepted, production will need this reference to link the reader to the Table.

Reviewers' comments:

Reviewer's Responses to Questions

**Comments to the Author**

1. Is the manuscript technically sound, and do the data support the conclusions?

Reviewer #1: No

Reviewer #2: Yes

2. Has the statistical analysis been performed appropriately and rigorously? 

Reviewer #1: No

Reviewer #2: Yes

3. Have the authors made all data underlying the findings in their manuscript fully available?

Reviewer #1: Yes

Reviewer #2: Yes

4. Is the manuscript presented in an intelligible fashion and written in standard English?

Reviewer #1: Yes

Reviewer #2: Yes

5. Review Comments to the Author

Reviewer #1: It is meaningful to elucidate the relationship between beta-glucanase and the availability of diets rich in beta-glucan.

However, in this manuscript, I think that only experimental data are only listed overall, and the results are not well organized as research papers. It is necessary to clarify the aim and the background of each experiment and revise the entire manuscript.

Terms should not be abbreviated unless necessary. In particular, do not abbreviate general terms consisting of two words.

Please proofread the entire manuscript carefully. The Tables are out of order and duplicates partly.

L116-119 The authors state that the objective of the current study was to investigate the effects under different housing environments and disease conditions, but it seems that the two experiments have not reproduce different housing environments and disease conditions (only vaccinated with a coccidia vaccine).

L126 What does “environmental pressures” mean?

L134 neonatal chicks? Please clarify the age of birds.

L138 The starting room temperature was …

→When the chicks placement, the room temperature was…

L150 Is it correct that the four treatments are the same as in Experiment 1? The treatment in Experiment 2 should be explained again.

L179 Coccidiosis challenge

→Coccidia vaccination

Vaccines are drugs that prevent the development of coccidiosis. It is not appropriate to interpret the changes caused by vaccination as coccidiosis.

L191 Rearing performance?

Please explain the reason why the composition of the feed used in Experiment 1 and Experiment 2 were different.

Materials and methods: To understanding easily, materials and methods for Experiment 1 and Experiment 2 should be described separately.

Experiment 1

Birds and Housing

Diets

Data collection

Sample collection

Experiment 2

Birds and Housing

Diets

Coccidia (not coccidiosis) challenge

Data collection

Sample collection

The results of Experiment 1 and Experiment 2 should be organized separately.

L254-255 Molar mass distribution curves were used to obtain β-glucan (Mp) ,.. ?

L262 ...by Zhao et al. with minor changes.

L306 -308 What is the interaction between main effects? Also, what do the value and degree of L307 mean? Please explain using suitable terms.

L317 Fig1 has not mention 33d. Please explain the reason in the materials and methods.

Reviewer #2: This work is very difficult to read due to the great detail of data (and of a certain repetition), as there are always two types of the same answer (SCFA μmol / g of wet ileal content Molar percentage of total SCFA, Content, Weight, Empty weight ( %) Length (cm / 100g) In addition, the use of expressions that must be corrected such as "soluble β-glucan weight average molecular weight" makes it even more difficult.

Interpretations about interactions should be reviewed, as there are some with errors. I suggest that the introduction and discussion be summarized and use a more objective language to make the article more attractive to the reader.

On the other hand, it is data-rich work.

6. PLOS authors have the option to publish the peer review history of their article (what does this mean?). If published, this will include your full peer review and any attached files.

Reviewer #1: **Yes: **takeshi kawasaki

Reviewer #2: No

---

## [Author Response · Author response to Decision Letter 0]

30 Dec 2020

We want to express our gratitude to the reviewers and editors for taking the time to volunteer their service to help peer-review this manuscript, and we believe that it will contribute to improving the manuscript. 

Editor

Line Comment

 Please include a copy of Table 10 which you refer to in your text on page 20.

Table 10 was included on page 33 (The table numbers were updated due to the changes in the experiment presentation order).

 We note you have included a table to which you do not refer in the text of your manuscript. Please ensure that you refer to Table 2 in your text

Table 2 was referred to in the text (L297 and L364).

Reviewer #1

The authors would like to thank the reviewer for taking time from 

Line Comment

 It is meaningful to elucidate the relationship between beta-glucanase and the availability of diets rich in beta-glucan.

However, in this manuscript, I think that only experimental data are only listed overall, and the results are not well organized as research papers. It is necessary to clarify the aim and the background of each experiment and revise the entire manuscript

The results were organized separately under each experiment in line with reviewer comments.

The aim and background of each experiment were described in the introduction (L93-103).

 Terms should not be abbreviated unless necessary. In particular, do not abbreviate general terms consisting of two words.

The following abbreviations were removed from the manuscript; BWG, FI, GC, NSP, TDF, IDF, SDF.

 Please proofread the entire manuscript carefully. The Tables are out of order and duplicates partly.

The entire manuscript was proofread.

The table numbers were updated due to the changes in the presentation of experiments in the entire manuscript.

L116-119 The authors state that the objective of the current study was to investigate the effects under different housing environments and disease conditions, but it seems that the two experiments have not reproduce different housing environments and disease conditions (only vaccinated with a coccidia vaccine).

Birds were housed in cages and on the floor in litter covered pens. These environments alter the nature of the exposure of birds to environmental microbes and the ability of birds to cycle organisms via coprophagy. The cage environment eliminates or greatly reduces coprophagy, and the environment is cleaner because of regular excreta removal. A major difference between the experiments was coccidiosis vaccination, with the degree of response exacerbated by high environmental humidity and litter moisture. The authors conclude that the housing environments and disease exposure are quite different in the two experiments

These differences were clarified in the introduction (L97-103).

L126 What does “environmental pressures” mean?

Deleted.

L134 neonatal chicks? Please clarify the age of birds.

Revised (L122 and L170).

L138 The starting room temperature was …

→When the chicks placement, the room temperature was…

The details about environmental conditions, including room temperature, were shorted as requested by Reviewer #2 (L123).

L150 Is it correct that the four treatments are the same as in Experiment 1? The treatment in Experiment 2 should be explained again.

Treatments for the two experiments were described separately in line with the reviewer comments (L127-131 and L 179-183).

L179 Coccidiosis challenge

→Coccidia vaccination

Vaccines are drugs that prevent the development of coccidiosis. It is not appropriate to interpret the changes caused by vaccination as coccidiosis.

Revised (L191).

L191 Rearing performance?

Revised (L139 and L202).

 Please explain the reason why the composition of the feed used in Experiment 1 and Experiment 2 were different.

Differences in feeds used in the experiments are related to logistical aspects of feed manufacturing and the animals' eventual use. 

In Experiment 1, small bird numbers meant that amounts of diet manufactured were small, and it would be challenging to manufacture two pelleted (crumbled) diets. Pelleting diets require more quantity to permit stabilization of pelleting conditions. The authors chose the grower diet, which best represents bird requirements over the entire trial. Experiment 1 birds were euthanized at the end of the trial and did not enter the human food chain. 

Because of the much larger numbers of birds in Experiment 2, it was possible to use both starter and grower diets. Birds from this experiment that were not used for tissue collection were marketed to a commercial processing facility. 

The Experiment 1 diet (grower) was almost identical to the Experiment 2 grower diet. The authors have not changed the manuscript in response to this reviewer's comment.

 Materials and methods: To understanding easily, materials and methods for Experiment 1 and Experiment 2 should be described separately.

Experiment 1

Birds and Housing

Diets

Data collection

Sample collection

Experiment 2

Birds and Housing

Diets

Coccidia (not coccidiosis) challenge

Data collection

Sample collection

Materials and methods for each experiment were separately described as requested by the reviewer.

 The results of Experiment 1 and Experiment 2 should be organized separately.

The results for the two experiments were organized separately under each experiment.

L254-255 Molar mass distribution curves were used to obtain β-glucan (Mp) ,.. ?

The parameters, including Mp, Mw and MW-10%, have been estimated by the software. Revised (L246-248).

L262 ...by Zhao et al. with minor changes.

Revised (L253).

L306 -308 What is the interaction between main effects? Also, what do the value and degree of L307 mean? Please explain using suitable terms.

The description of the interaction was modified for clarity (L295-297).

L317 Fig1 has not mention 33d. Please explain the reason in the materials and methods.

The same distribution patterns were observed for day 33, but only d 11 was presented as an example. This was explained in the results (L374-376).

Reviewer #2

Line Comment

 This work is very difficult to read due to the great detail of data (and of a certain repetition), as there are always two types of the same answer (SCFA μmol / g of wet ileal content Molar percentage of total SCFA, Content, Weight, Empty weight ( %) Length (cm / 100g) In addition, the use of expressions that must be corrected such as "soluble β-glucan weight average molecular weight" makes it even more difficult.

The authors have used the correct and standard technical terms to mention the analyzed parameters.

Please note that the presentation order of the sections ‘materials and methods’ and ‘results’ were changed entirely since reviewer #1 has requested to organize Experiment 1 and Experiment 2 separately for these two sections. Therefore, the order of the tables and line numbers were also changed.

 Interpretations about interactions should be reviewed, as there are some with errors.

The interactions were reviewed, and the changes were made when interpreting some of the interactions as mentioned below under reviewer comments (L334-336, L407-408, L501-504). 

 I suggest that the introduction and discussion be summarized and use a more objective language to make the article more attractive to the reader.

On the other hand, it is data-rich work

The introduction and discussion were shortened and revised (The line numbers range for the respective paragraph is mentioned within brackets).

Introduction;

The first paragraph was shortened (L52-58).

In the second paragraph, the details about fructo-oligosaccharides and mannan-oligosaccharides were removed (L59-73).

The last sentence of the third paragraph was removed (L74-84).

Discussion;

The statement regarding the first time reporting the molecular weight data in the literature was removed (L552-573).

The sentence regarding the gut microbial population in pigs and the details about NSP characteristics and viscosity was removed (L574-587).

The discussion about feed passage time was deleted (L588-605).

The histo-morphology section was shortened (L643-653). 

The discussion about butyric acid was eliminated (L667-680).

PONE-D-20-20511

 the tittle is enormous and I cannot understand what is mileal digesta soluble β-glucan molecular weightw

The title was not revised because the authors believe it is clear and descriptive.

 Coccidiose vaccine is not the same as coccidiose challenged birds, the dose, the way, many elements are different from a challenge in the field. So I suggest changing the tittle and wherever it is discussed

Changed as coccidiosis vaccination through-out the manuscript, including the title (L 6, 34, 47, 100, 113, 191, 192, 686 and 697).

 Abstract:

I cannot understand what is “the soluble β-glucan weight average molecular weight (Mw)” It is impossible to understand what is peak molecular weigh without reading the M&M. Need changing

Mw was changed as ‘weighted average molecular weight’ in the entire manuscript (L36, 247, 249, and Table 2). The terms ‘weight average’ and ‘weighted average’ molecular weight are used in human nutrition research, and the definition of both terms is the same. The weight/weighted average differs from the number average because the carbohydrate chain’s weight fraction is considered when determining the weighted average. Peak molecular weight was not changed since it is a standard term. The authors believe it is impossible to define them in the abstract because of length, and the definitions were included in the materials and methods with a reference (L249-251). 

L38 smallest 10% β-glucan molecule (MW-10%) was lower with BGase.

It was revised as ‘BGase addition’. The definition of MW-10% was not changed, as the authors believe it is clear and descriptive (L39).

L76 I’d like to understand why the decrease in of molecular weight of arabinoxylan is the cause of the decrease and increase of pathogenic/beneficial bacterias

 In addition, exogenous xylanase in wheat-based diets increased

 the number of gastro-intestinal beneficial bacteria, including lactic acid bacteria, while reducing pathogenic bacteria in broiler chickens [20,21], probably by decreasing the molecular weight of soluble arabinoxylan derived from the wheat.

The reduction of arabinoxylan MW increases beneficial bacteria and is attributed to the increased access of these substrates into the caeca because caecal entry is restricted to small, soluble, and less viscous material, or it can be due to the microbial preference for utilizing small MW carbohydrates over the large MW material. This was not included in the introduction since it will further increase the length.

 Introduction- too long, please you need to be more concise

Introduction was shortened (paragraph line numbers range is included within brackets).

The first paragraph was shortened (L52-58).

In the second paragraph, the details about fructo-oligosaccharides and mannan-oligosaccharides were removed (L59-73).

The last sentence of the third paragraph was removed (L74-84).

L101-104 The primary mechanism is generally accepted to be a positive modulation of the diversity and relative abundance of bacteria in the digestive tract microbial community, and thereby the control of enteric disease and stimulation of immune function in broiler chickens [29-31].

I disagree with you. The new theories tell about anti-inflamatory activicty (see Niewold, T. 2007. The nonantibiotic anti-inflammatory effect of antimicrobial growth promoters, the real mode of action? A hypothesis. Poult. Sci. 86:605–609.)

The theory about the anti-inflammatory activity was also mentioned (L89-90).

L117 ileal digesta soluble β-glucan molecular weight distribution (very confuse)

It is long, but the authors believe it is informative and is necessary to describe the specific parameter measured in the current study. No changes were made.

L122 This should result in improved performance of broiler chickens and

Revised (L109-110).

L135 The dimensions of the cages were 51 cm in length, 51 cm in width

and 46 cm in height. The grid size of the wire mesh floor of each cage was 2.54 × 2.54 but was covered by a 1.27 × 1.27 cm mesh until d 7. There were two levels of battery cages that were in two rows with back to back cages. The starting 138 room temperature was 32°C, and it was gradually decreased by 2.8°C per week. The minimum light intensity was 25 lux during the experimental period,

Line 141-144 Each cage had a front-mounted feed trough (51 cm in length) and two height-adjustable nipple drinkers. Extra feed and water were supplied to the birds from d 0 to 5 using supplementary chick feeders (50 cm long, plastic) and ice cube trays (16 cells), respectively.

Why all this superfluos details? Is is a paper with few pages. It´s not the entire thesis. I’ve never seen so many specifications.

The paragraph was shortened (L121-124). 

 Experiment 2- how many birds per m square?

The flock density was added (L171-172).

L153 Additional feed and water were supplied to each pen using a

cardboard egg tray and an ice cube tray, respectively, for the first week. Straw was placed in each room at a thickness of 7.5-10 cm. The room temperature was 33°C at the beginning of the experiment and was gradually reduced to 21°C by d 25. Day length was gradually reduced from 23 h at d 0 to 17 h at d 12, and the light intensity was set to 20 lux at the start of the experiment and gradually decreased to 10 lux by d 10.

No need of such detail

Just say” The feed supply and water were ad libitum, and the animals were kept in thermal confort and day length was gradually reduced from 23 h at d 0 to 17 h at d 12.”

The paragraphs were shortened (L121-124 and L174-177).

L163 (Bacitracin- Zoetis Canada Inc.,Kirkland, QC, Canada)

Revised (L130).

L164 t 4.4 mg/kg and Salinomycin Sodium (Phibro Animal Health

165 Corporation, Teaneck, NJ- at 25 mg/kg). two parenthesis do not exist

Revised (L130-131).

L187 ingestion by the birds. In addition, 60% relative humidity was maintained in the rooms, , to facilitate oocyst cycling.

Revised (L199-200). 

L224 80°C for β-glucan molecular weight analysis

Revised (L165-166).

L248 Beta-glucan molecular weight distribution- why the word distribution? Just to confuse the reader

The word ‘distribution’ was removed through-out the paper.

L255 weight average molecular weight (Mw) - what is that? Average molecular weight I understand, but weigh average weight no.

The weighted average emphasizes the weight fraction of the molecule/carbohydrate chain. A weighted average is more accurate than a number average because all numbers in a data set are assigned an identical weight. The definition was included in the materials and methods (L249-251).

Number average molecular weight = ∑ NiMi/ ∑ Ni

Weight or weighted average molecular weight = ∑ NiMi2/ ∑ NiMi

Mi is the weight of the molecule i

Ni is the number of molecules of weight M

L258 Average molecular weight is the average of the molecular weights of all β-glucan

Revised (L250). 

L262 ...by Zhao et al. with minor changes.

Revised (L253).

L262 The internal standard for the analysis was made up of 20 ml of 25%. What means internal standard?

The internal standard is a known concentration of a substance present in every sample that is analyzed. Internal standards are commonly used in chromatography.

L299 8.70%, respectively, and the same fractionswere 15.2, 13.7, 1.6 and 0.68%,

Revised (L291).

L306-310 Look Table 2 – Mp was not different between 0 Med x 0,1 Med. I don´t see any interaction in the sense the paper is searching. Only for Mp there is an interaction

Interactions are found for Mp in both experiments (L295-297 and L363-367). Please check the P values and mean separation letters in Table 2. No changes were made.

L311 In Experiment 2, interactions were found for all molecular weight criteria at both ages (11 and 33 d) except for Mw at 11 d, which was also unaffected by medication or BGase. No interaction of interes was found for MW-10% ( 33322 a= 26065 a and 7250 b = 1058b) So, no interaction!

Interactions are found for MW-10% at both ages (L 366-368). Please see the P values and mean separation in Table 2. No changes were made.

L305 Beta-glucan molecular weight 

Only the figures are “distribution”. Avoid this term, except for the figures 1A ,1B, 1C

The word ‘distribution’ was removed.

L320 shown by curve placement relative to the blue line at x-axis point 1e4.

What is point 1e4 ?

1e4 is a random point that is selected to compare the three graphs since the X-axis is different in the graphs. It was mentioned in the text (L372-373).

L330 Ileal digesta viscosity was not affected by medication in Experiment 1, but was reduced

Revised (L303).

L331 with the use of BGase. In Experiment 2 at 11 d, an interaction was found between medication and BGase; BGase reduced viscosity without dietary medication. In the interaction, the highest viscosity was noted for the treatment without medication or BGase, and the lowest was the treatments with BGase; treatment with medication and without BGase was intermediate. At d 33 in Experiment 2, BGase decreased viscosity, but there was no medication effect

Medication did not decrease significantly the viscosity (9,73a x 6.04 ab) . It decreased numerically, but not statistically . “a and ab” are not different. This is the interaction that counts

Only the enzyme had an important effect. Please change de sentence

The authors agree that ‘a’ and ‘ab’ are not statistically significant, which is why it was stated as ‘BGase reduced viscosity without dietary medication’ (L382-383). The authors have not mentioned ‘medication decreasing the viscosity at d 11’. No changes were made.

L344 … were also not affected by treatment (Table 5). Noteworthy, the interaction between medication and BGase tended to be significant (P = 0.06-0.09) for the concentrations of total and individual SCFA.

Like I said before, you cannot guarantee that the major difference is between 0 Bgase (284) x med (267). Probably the difference appears between 310 and 267 . So reevaluate your analysis to know exactly why there is an interaction. You need a lsmeans analysis to see that

The authors have just mentioned that interaction tended to be significant based on the P values and did not mention the difference between specific means (L313-314). No changes were made.

 Table 4 jumps to Table 6

All the table numbers were updated since the presentation of results changed in the entire manuscript.

L348-358 the same thing only valeric has an important interaction (table 6)

The authors have mentioned the interaction is found only for valeric acid, and the other mentioned interactions trended to be significant. No changes were made.

L364 The molar percentages isobutyric acids were decreased by medication, while enzyme use decreased and increased the proportions of acetic and butyric acids, and valeric acid, respectively. (???)

For Propionic it´s not true: pro – 0 Med ( 49,7b) = Med (36,4 bc)

You cannot use “respectively” comparing two categories and 3 things!!!

Revised (L403-405).

L366 The interaction between main effects was significant for the proportional isovaleric levels, with enzyme tending to decrease levels in unmedicated diets and increase levels in medicated diets. Although the above effects were significant, differences were small.

Here you used “tending” for a P<0.03 and explains the difference is small??? You are tending your interpretations. No tendency here. This is a significant interaction. In this case medication decreased isov in the absence of the enzyme and the enzyme improved isov only without medication

The authors agree that it is a significant interaction, and the changes were made in the text (L407-408).

L430 content weights of the gizzard, jejunum and small intestine. The gizzard content weight to be higher and lower with enzyme use in birds fed non-medicated and medicated diets, respectively.

You are using in wrong sense the word “tendency”. Now-a-days it´s been used for Probabilities between 0.05 and 0.10, not because a mean is similar statistically, although greater or minor numerically (a x ab). In this case, gizzard content without med and enzyme is lower than with med. Only that.

The authors agree with the argument, and the changes were made in the text (L334-336).

L497 … increased gain, while enzyme did not affect gain in birds fed diets without medication and increase gain in the medicated diet. Weight gain from 11 to 22 d was increased by enzyme regardless of diet medication. But med alone also increased WG.

Revised (L501-504).

L506 Interactions were found between medication and BGase for F:G in all periods.

Medication feed efficiency throughout the trial, but as was the case for BWG, the nature of the interaction with enzyme use changed with bird age. During the 0-11 d period, F:G increased with enzyme use when birds were fed non-medicated diets, but had no effect when the medication was used. For the remainder of the periods, including the total trial, enzyme F:G in birds fed non-medicated diets, but had no effect in broilers consuming medicated diets.

For this response prefer the the words “worse or improve” F:G instead “increase or decrease”, because the nature of the response ( the higher F:G , the worse result)

Changed (L514-518).

L520 translocation. It can be concluded that the vaccination with Coccivac-B52 induced a disease 521 challenge in the birds from Experiment 2 according to the detailed analysis of mortality data.

Are you not overreacting about this data? The mortality between the two trials was practicaly the same, one with vaccination,the other without vaccination. Besides, ONLY 4,3 % in Exp 2 was due to necrotic enteritis. How do you know that in Exp 1 there was not a cause related to systemic infection, since you did not analyze intestine? For me, you cannot conclude that.

The authors agree with the points made by the reviewer and removed the statement about the disease challenge.

L530 With minor exceptions, all three molecular weight responsesfor soluble ileal digesta β-

Parameter has other meaning- A parameter is a measurable characteristic of a or a population, such as a mean or a standard deviation …

Changed it as ‘molecular weight responses’ (L531).

L536 glucan depolymerization is independent of the and the age of the animal. Further,

You cannot say chickens were sick. Vaccination and sickness are completely different things

The ‘disease status and age’ was changed as ‘vaccination status’ (L537).

L536 glucan depolymerization is independent of the disease status and the age of the animal. Further, Mw from Experiment 1 also supports the reduction of molecular weight in the ileal digesta soluble β-glucan with the use of BGase.

What is the hypothesis for this sentence? Why Exp 2 would not support the effectiveness of the enzyme? I suggest taking this out

In Experiment 2, Mw increased with the enzyme at d 33 (and numerically at d 11) in the medicated treatments. It was mentioned in the next sentences, and a sentence was added, mentioning the possible reasons (L543-545).

L538 In contrast, BGase increased Mw in Experiment 2 in the treatments with antibiotics.

That´s not true. At 11 days, no significant differences

It was changed as ‘at d 33’ instead of ‘both days’ (L539-540).

L542 The reduction of β-glucan molecular weight and the increased proportion of

 small molecular weight soluble β-glucan encourage the assessment of performance and digestive tract characteristics through increased β-glucan fermentation in broilers.

Take out this sentence. No important information is in this sentence.

The authors are of the opinion that the sentence connects molecular weight results with other responses in the research. Therefore, the sentence was not removed, but it was revised to clarify the concept (L545-548) hopefully.

L544 Further, the proportion of small molecular weight β-glucan might be a critical assessment in chickens since chicken ???? 

Why might be a critical assessment ? For me it is a good response to improve fermentation.

Revised (L548-549).

L551 utilize fibre [46]. Further, molecular weight parameters were lower in Experiment 1 compared to both ages in Experiment 2.

Since in exp 1, birds were close in age with the second age of Exp 2, how do you interprete this result? Don´t you think medication could be influencing, “protecting” de B glucan of fermenting?

The same medication has been used in both experiments and could not affect molecular weight differences in the two experiments. The authors have mentioned the possible reasons in the text (L560-573). No changes were made.

L569 … support the difference related to the microbial enzyme activity [48]. To the best of our knowledge, this, , isthe first to document molecular weight. Other researches not published are not valid in this terms.

The sentence was deleted.

L572 …BGase effect on the reduction of ileal β-glucan molecular weight in this study is in…

Revised (L572).

L574 The molecular weight responses in the two experiments decreased with medication

Revised (L574).

L585 It demonstrates the higher efficacy of feed BGase in comparison to the BGase originated from microbiota in the chicken gastro-intestinal tract of degrading high molecular weight β-glucan.

What does this sentence mean? I cannot understand

The sentence was removed.

L591 ileal viscosity in broiler chickens, you’ve just said that. Don’t need repeat.

Revised (L584-585).

L593 Nevertheless, viscosity changes at 11 d are similar to the molecular weight data where enzyme decreased values, but the decrease was smaller with medicated diets, and it is primarily due to the low viscosity in the treatments with medication when BGase was not used.

This is wrong. Means are not different between 9.73 a x 6.04 ab

The sentence was deleted.

L598 …digesta [56-58]. Further, medication might have shifted the ileal microbial population in a way that leads to increased intestinal mucus production, which can contribute to ileal viscosity [59,60]. In addition, high amounts of NSP in the diet also increase intestinal mucus production in monogastric animals [59,61].

Lot of explanation for no reason. Take out

Deleted. 

L602 Overall, BGase reduced the empty weights, lengths, and content weights in the digestive tract segments in both experiments, which agrees with previous broiler research that used the same diets but without medication [62].

I don´t agree with you. Some segments were reduced, but not some response so strong that can support this sentence. Modify the sentence , please.

For example, at 11 days (exp 2), SI increased with the enzyme use.

Revised (L588).

L608 … thereby increases digestive function in the broiler chickens [67,68]. Further, HB mediated larger digestive tract might hold more digesta that leads to increased gastro-intestinal content weights in the current study.

Please, show me where there are consistent increase in gastro-intestinal content weights in the current study??

The sentence was deleted.

L605 … efficiency associated with enzyme use and has been reported previously [63,64]. In addition, the reduction of gastro-intestinal content weights might be associated with increased feed passage rate in the gastro-intestinal tract [65,66]

In your experiment FI was not affected. Increased feed passage rate would increase FI, don´t you think? 

In my opinion you are trying to exacerbate same small effects with this long discussion without a data that supports your theory

The sentence was deleted.

L624 contain BGase since the enzyme also decreases digestive tract size by increasing nutrient digestibility in broiler chickens.

Cannot affirm that, You did not do digestibility of nutrients. Bglucanase did not improve any performance in EXP 1, don’t forget. It actually, tended to DECREASE WG and more, with medication, B glucanase WAS NOT effective in improving BWG, ie, medication has a strong effected than the enzyme

Removed this part of the sentence - ‘by increasing nutrient digestibility in broiler chickens’.

L627 BGase and antibiotics on carbohydrate fermentation. Diet BGase and medication depolymerized soluble β-glucan in HB in the ileal digesta of broiler chickens, which may influence carbohydrate fermentation in the lower digestive tract.

You’ve already said that; summarize, please

Revised (L606-608).

L705 Too long this discussion . The intestinal structute results do not show enough news to justify so many discussion

This part of the discussion was shortened (L643-653).

L725 efficiency in the birds aged < 11 d but increased these responses after d 11. Moreover, BGase

Revised (L681-682).

L735 medication and enzyme on production responses in the battery cage study.

Revised (L698-699).

---

## [Decision Letter · Decision Letter 1]

8 Feb 2021

PONE-D-20-20511R1

Effects of exogenous β-glucanase on ileal digesta soluble β-glucan molecular weight, digestive tract characteristics, and performance of coccidiosis vaccinated broiler chickens fed hulless barley-based diets with and without medication

PLOS ONE

Dear Dr. Newkirk,

Thank you for submitting your manuscript to PLOS ONE. After careful consideration, we feel that it has merit but does not fully meet PLOS ONE’s publication criteria as it currently stands. Therefore, we invite you to submit a revised version of the manuscript that addresses the points raised during the review process.

Minor thoughts are attached for your consideration. Careful editing is still needed.

We look forward to receiving your revised manuscript.

Kind regards,

Arda Yildirim, Ph.D.

Academic Editor

PLOS ONE

Additional Editor Comments (if provided):

I feel that the manuscript is dealing with a good topic but lacks in the quality of preparation. Please review the referee comments (reviewer#1 and 2) and make your minor revision. Thanks for your hard work.

Reviewers' comments:

Reviewer's Responses to Questions

**Comments to the Author**

1. If the authors have adequately addressed your comments raised in a previous round of review and you feel that this manuscript is now acceptable for publication, you may indicate that here to bypass the “Comments to the Author” section, enter your conflict of interest statement in the “Confidential to Editor” section, and submit your "Accept" recommendation.

Reviewer #1: (No Response)

Reviewer #2: (No Response)

2. Is the manuscript technically sound, and do the data support the conclusions?

Reviewer #1: Yes

Reviewer #2: Partly

3. Has the statistical analysis been performed appropriately and rigorously? 

Reviewer #1: Yes

Reviewer #2: No

4. Have the authors made all data underlying the findings in their manuscript fully available?

Reviewer #1: Yes

Reviewer #2: Yes

5. Is the manuscript presented in an intelligible fashion and written in standard English?

Reviewer #1: Yes

Reviewer #2: Yes

6. Review Comments to the Author

Reviewer #1: Thank you for correcting the overall structure of the manuscript and the points pointed out.

This is a treatise with a large amount of data submitted. In the abstract, it is recommended that you briefly summarize the points in the order of Experiment 1, Experiment 2, and Conclusion to make the content easier for the reader to understand.

L30-

Broilers were…

→In the present study, broilers were…

L32-

The reviewer recommends correcting the below sentences as follows.

"In Experiment 1, 160 broilers were assigned to 10 cages from day 0 to day 28; we investigated the effects of BGase treatment and the effects of medication. The soluble β-glucan weighted average molecular weight (Mw) in the ileal digesta was … for the smallest 10% β-glucan (MW-10%) was lower with BGase addition."

L33-L35

The reviewer recommends the sentences of L33-L35(“In Experiment 2, broilers (2376) were housed in…in each of nine rooms.”) are moved to the next location of the Experiment 1.

In the main manuscript, please consider the following points.

L98-L103

The sentences of “Experiment 1 was completed in … and microbial exposure.” explain to perform in the current study. The reviewer recommends the sentences move to the next location of L104-107.

L414, L421, L500

What is "main effect"? The authors should need to present them specifically.

or, for example, "No effect of the interactions of BGase and medications were found for …"

Reviewer #2: The article improved with the corrections made and it is interesting as I already said, although very large. However, I do not understand that it should be published in this way, as there is a wrong interpretation of the meaning of the interactions. In statistics, when there are interactions, it is necessary to know how to interpret them. The fact that ANOVA shows that there is an interaction between two or more factors is not enough for the interaction to be discussed in order to prove the hypothesis of a work, since the means test must also show the same. The authors are studying the interaction between enzyme and medication. The interaction that counts is whether the enzyme acts in way A with the drug or acts in way B without the drug. This never happened (and the averaging test clearly shows). I'll give you an example: in table 2, Mp (Exp 1) was higher with medication and lower with enzymes. Enzymes + Medication did not change this fact, since the Mp is numerically higher (10401) than without medication (7793), both, by means test are equal (same letter). Therefore, it is not possible to state that using enzymes alone, decreases Mp more than using an enzyme with medication.

One solution would be to use a less demanding means test, in which case increasing the chance of a Type I error.

However, with the current statistics, my vote is for rejection.

7. PLOS authors have the option to publish the peer review history of their article (what does this mean?). If published, this will include your full peer review and any attached files.

Reviewer #1: **Yes: **Takeshi Kawasaki

Reviewer #2: **Yes: **ANDREA MACHADO LEAL RIBEIRO

---

## [Author Response · Author response to Decision Letter 1]

12 Mar 2021

The authors would like to thank the editor and reviewers for taking time to review the manuscript and for the comments.

Reviewer #1

Line Comment

 Thank you for correcting the overall structure of the manuscript and the points pointed out.

This is a treatise with a large amount of data submitted. In the abstract, it is recommended that you briefly summarize the points in the order of Experiment 1, Experiment 2, and Conclusion to make the content easier for the reader to understand.

The abstract was organized in the order of Experiment, 1, Experiment 2 and then conclusion (L 33-45).

L 30 Broilers were…

→In the present study, broilers were…

Revised (L 30).

L 32 The reviewer recommends correcting the below sentences as follows.

"In Experiment 1, 160 broilers were assigned to 10 cages from day 0 to day 28; we investigated the effects of BGase treatment and the effects of medication. The soluble β-glucan weighted average molecular weight (Mw) in the ileal digesta was … for the smallest 10% β-glucan (MW-10%) was lower with BGase addition."

The sentences were revised (L 33-34).

L 33-35 The reviewer recommends the sentences of L33-L35(“In Experiment 2, broilers (2376) were housed in…in each of nine rooms.”) are moved to the next location of the Experiment 1.

The sentences were moved (L 37-39).

L 98-103 The sentences of “Experiment 1 was completed in … and microbial exposure.” explain to perform in the current study. The reviewer recommends the sentences move to the next location of L104-107.

The sentences were moved as requested (L 103-108).

L 414, 421, 500 What is "main effect"? The authors should need to present them specifically.

or, for example, "No effect of the interactions of BGase and medications were found for …"

Main effects were clarified in the sentences (L 337, 405, 420, 429, 437, 487, 520).

Reviewer #2

Line Comment 

 The article improved with the corrections made and it is interesting as I already said, although very large. However, I do not understand that it should be published in this way, as there is a wrong interpretation of the meaning of the interactions. In statistics, when there are interactions, it is necessary to know how to interpret them. The fact that ANOVA shows that there is an interaction between two or more factors is not enough for the interaction to be discussed in order to prove the hypothesis of a work, since the means test must also show the same. The authors are studying the interaction between enzyme and medication. The interaction that counts is whether the enzyme acts in way A with the drug or acts in way B without the drug. This never happened (and the averaging test clearly shows). I'll give you an example: in table 2, Mp (Exp 1) was higher with medication and lower with enzymes. Enzymes + Medication did not change this fact, since the Mp is numerically higher (10401) than without medication (7793), both, by means test are equal (same letter). Therefore, it is not possible to state that using enzymes alone, decreases Mp more than using an enzyme with medication.

One solution would be to use a less demanding means test, in which case increasing the chance of a Type I error.

However, with the current statistics, my vote is for rejection

The authors reviewed all the interaction interpretations in the results section (tables and text) and the necessary changes were done in the wording to clarify the interactions according to mean separation (L 297-303, 334-335, 341-342, 372-376, 393-395, 409-412, 422-423, 469-471, 488-489, 531-532).

PONE-D-20-20511

Line Comment 

L 152, 160-161 … Total ileal and caecal contents were collected into plastic centrifuge tubes and stored at -20°C for the analysis of SCFA.

… Total ileal contents were collected into plastic snap-cap vials (pooled from all the birds in a cage) and centrifuged for 5 min at 17013 × g using a Beckman microfuge (Model E 162 348720, Beckman Instruments, INC, Palo Alto, CA).

How it is possible? Two total ileal contents collected in different places? Different birds?

There were 4 birds in each cage. Two birds per cage was used for pH and SCFA sample collections. Total ileal content from each bird was collected to a tray and only a portion of it was added to the centrifuge tube after mixing. The rest of the ileal content from these two birds was put into the plastic snap cap vial for centrifugation. Another two birds per cage were used for digestive tract morphology data, and the ileal contents from these two birds were also added to the same snap-cap vial. So, ileal contents from all 4 birds per cage were pooled before centrifugation and used for viscosity and molecular weight analyses. The sentences were revised for a clearer explanation (L 154-156, 162-163). 

L 149, 153 …. Two birds were used

…. Two

Birds were used to collect ...

The same two birds? If are the same do not repeat. If they are different, say it.

There were 4 birds in each cage. Two birds were used for pH and SCFA sample collection. Another two birds were used for digestive tract morphology data collection. They are different. The sentence was revised (L 156).

L 364 for Mw at 11 d, which was also unaffected by medication or BGase (Table 2).

Why also?

Deleted the word ‘also’ (L 371).

---

## [Decision Letter · Decision Letter 2]

9 Apr 2021

PONE-D-20-20511R2

Effects of exogenous β-glucanase on ileal digesta soluble β-glucan molecular weight, digestive tract characteristics, and performance of coccidiosis vaccinated broiler chickens fed hulless barley-based diets with and without medication

PLOS ONE

Dear Dr. Newkirk,

Thank you for submitting your manuscript to PLOS ONE. After careful consideration, we feel that it has merit but does not fully meet PLOS ONE’s publication criteria as it currently stands. Therefore, we invite you to submit a revised version of the manuscript that addresses the points raised during the review process.

Minor concerns are attached for your consideration. Please make careful your peer revision.

We look forward to receiving your revised manuscript.

Kind regards,

Arda Yildirim, Ph.D.

Academic Editor

PLOS ONE

Journal Requirements:

Additional Editor Comments (if provided):

Minor concerns are attached for your consideration. Please careful editing is still needed. Thanks.

Reviewers' comments:

Reviewer's Responses to Questions

**Comments to the Author**

1. If the authors have adequately addressed your comments raised in a previous round of review and you feel that this manuscript is now acceptable for publication, you may indicate that here to bypass the “Comments to the Author” section, enter your conflict of interest statement in the “Confidential to Editor” section, and submit your "Accept" recommendation.

Reviewer #1: (No Response)

2. Is the manuscript technically sound, and do the data support the conclusions?

Reviewer #1: Partly

3. Has the statistical analysis been performed appropriately and rigorously? 

Reviewer #1: Yes

4. Have the authors made all data underlying the findings in their manuscript fully available?

Reviewer #1: Yes

5. Is the manuscript presented in an intelligible fashion and written in standard English?

Reviewer #1: Yes

6. Review Comments to the Author

Reviewer #1: I could understand the manuscript's overall content, but there are still some minor points that need to be corrected. Please attention to the following:

L26-48: Abstract; It may be easier for the reader to understand the content if the abstract is divided into items as shown below.

Limited use of medication in poultry feed led to the investigation of exogenous enzymes as antibiotic alternatives for controlling enteric disease. The objective of this study was to evaluate the effects of diet β-glucanase (BGase) and medication on β- glucan depolymerization, digestive tract characteristics, and performance of broilers.

Materials and methods: Broilers were fed hulless barley (HB) based diets with BG ase (Econase GT 200P from AB Vista; 0 and 0.1%) and medication (Bacitracin and Salinomycin Na; with and without) arranged as a 2 × 2 factorial. In Experiment 1, 160 broilers were housed in cages from d 0 to 28. Each treatment was assigned to 10 cages. In Experiment 2, broilers (2376) were housed in floor pens and vaccinated for coccidiosis on d 5. Each treatment was assigned to one floor pen in each of nine rooms.

Results: In Experiment 1, the soluble β-glucan weighted average molecular weight (Mw) in the ileal digesta was lower with medication in the 0% BGase treatments. Peak molecular weight (Mp) and Mw were lower with BGase regardless of medication. The maximum molecular weight for the smallest 10% β-glucan (MW-10%) was lower with BGase addition. In Experiment 2, Mp was lower with medication in 0% BGase treatments. Beta-glucanase resulted in lower Mp regardless of medication, and the degree of response was lower with medication. The MW-10% was lower with BGase despite antibiotic addition. Body weight gain and feed efficiency were higher with medication regardless of BGase use through-out the trial (except d 11-22 feed efficiency). Beta-glucanase resulted in higher body weight gain after d 11 and worsened and improved feed efficiency before and after d 11, respectively, in unmedicated treatments.

Conclusion: BGase and medication caused the depolymerization of soluble ileal β-glucan. Beta-glucanase acted as a partial replacement for diet medication by increasing performance in coccidiosis vaccinated broilers.

L30, L47, L350, Table9, L519, Table20, L665, L687, L688, L703, L707, L710, L711, L714, and L728: "Performance," which is often used throughout, is an ambiguous expression that hinders the content's objective understanding. They should specifically indicate "bodyweight," "feed intake," etc., respectively.

L104-114: The content of the individual experiments and the results of conducting them are objectively presented in the main contents of the present study. L104-114, which is an inference of subjective and concreted experimental results, should be deleted.

L305, Table3, L391: “Viscosity” should be replaced with “The viscosity of ileal supernatant” indicated by the material and method. Please unify the terms.

L478: The weight of the contents of the small intestine is not the result of morphological observation of the digestive tract, but the result of "measurement of the contents of the small intestine". Heading titles must be separated.

L483: As with the L478, the headings need to be separated.

L467: L467-502; Alternatively, it may be appropriate to say, for example, "digestive organ morphology and weight of contents" instead of "digestive tract morphology."

L604-605: I think "ileal viscosity" is probably not an accurate expression. Is the ileum flexible? Is it the viscosity of the gastrointestinal contents (ileal supernatant)?

L691-692: What does growth performance mean? What is the positive influence?

L687, L714: I don't think "performance variables" and "production cycle" are appropriate words to use in this context. Specific items used as performance indicators should be shown individually.

Also, the term production cycle used here seems to refer to the rearing period, inferring from the context.

Overall, please pay special attention to the terms' appropriateness and the unity of the terms, check and correct them again.

7. PLOS authors have the option to publish the peer review history of their article (what does this mean?). If published, this will include your full peer review and any attached files.

Reviewer #1: **Yes: **Takeshi Kawasaki

---

## [Author Response · Author response to Decision Letter 2]

12 Apr 2021

The authors would like to thank the editor and reviewer for taking time to review the manuscript and for the comments.

The following changes were made to the reference list.

• DOI was inserted/revised in the following references.

50. Donaldson GP, Lee Sm, Mazmanian SK. Gut biogeography of the bacterial microbiota. Nat Rev Microbiol. 2016;14: 20–32. doi:10.1038/nrmicro3552

52. Croom J, Chichlowski M, Froetschel M, McBride BW, Qui R, Koci MD. The effects of direct-fed microbial, primalac, or salinomycin supplementation on intestinal lactate isomers and cecal volatile fatty acid concentrations in broilers. Int J Poult Sci. 2009;8: 128–132. doi:10.3923/IJPS.2009.128.132

56. Rada V, Marounek M. Effect of monensin on the crop microflora of broiler chickens. Ann Zootech. 1996;45: 283–288. doi:10.1051/animres:19960308

67. Elwinger K, Berndtson E, Engström B, Fossum O, Waldenstedt L. Effect of antibiotic growth promoters and anticoccidials on growth of Clostridium perfringens in the caeca and on performance of broiler chickens. Acta Vet Scand. 1998;39: 433–441. doi:10.1186/BF03547769

• The references were changed for the reference list numbers 40 and 45.

40. Karunaratne ND, Classen HL, Ames NP, Bedford MR, Newkirk RW. Effects of hulless barley and exogenous beta-glucanase levels on ileal digesta soluble beta-glucan molecular weight, digestive tract characteristics, and performance of broiler chickens. Poult Sci. 2021;100: 100967. doi:10.1016/j.psj.2020.12.064

45. Karunaratne ND, Newkirk RW, Van Kessel AG, Bedford MR, Classen HL. Hulless barley and beta-glucanase levels in the diet affect the performance of coccidiosis-challenged broiler chickens in an age-dependent manner. Poult Sci. 2021;100: 776–787. doi:10.1016/j.psj.2020.10.036

• The reference number 69 was removed, and the text was updated.

69. Karunaratne ND, Bedford MR, Newkirk RW, Classen HL. Graded levels of hulless barley and β-glucanase affect the performance of broiler chickens vaccinated for coccidiosis in an age dependent manner. Poult Sci. 2017b;96 (E-suppl. 1): 50.

Reviewer #1

Line Comment 

L26-48 Abstract; It may be easier for the reader to understand the content if the abstract is divided into items as shown below.

The abstract was divided into items as requested (L 27-49).

L30, L47, L350, Table9, L519, Table20, L665, L687, L688, L703, L707, L710, L711, L714, and L728

 "Performance," which is often used throughout, is an ambiguous expression that hinders the content's objective understanding. They should specifically indicate "bodyweight," "feed intake," etc., respectively.

Revised using the terms ‘body weight gain’, ‘feed intake’ and ‘feed to gain ratio/feed efficiency’ (L 353, 523, 669, 691, 708, 712, 715, 716, 724, 733-734).

Changed as ‘growth performance’ because the length of the abstract increase if use all the terms (L 30, 48).

L104-114 The content of the individual experiments and the results of conducting them are objectively presented in the main contents of the present study. L104-114, which is an inference of subjective and concreted experimental results, should be deleted.

This statement was included in the objective and hypothesis section to clearly emphasis the reason for conducting two experiments in the current study as requested by Reviewer #2 in the previous reviewing process (L 104-115). No changes were made.

L305, Table3, L391

 “Viscosity” should be replaced with “The viscosity of ileal supernatant” indicated by the material and method. Please unify the terms.

Revised as “The viscosity of ileal supernatant’ (L 306, Table 3, 394).

L478 The weight of the contents of the small intestine is not the result of morphological observation of the digestive tract, but the result of "measurement of the contents of the small intestine". Heading titles must be separated.

A separate heading title was added for the content weights as “measurements of the contents of the digestive tract’ (L 339). 

L483 As with the L478, the headings need to be separated.

A separate heading title was added for the content weights as “measurements of the contents of the digestive tract, and digestive organ morphology’ (L 491-492). The table numbers 17 and 18 were switched because of the changes of the content weight paragraphs (L 483, 494).

L467: L467-502

 Alternatively, it may be appropriate to say, for example, "digestive organ morphology and weight of contents" instead of "digestive tract morphology."

The digestive tract morphology and content weights sections were separated, and separate heading titles were added as mentioned in the previous two comments (L 339, 491-492).

L604-605 I think "ileal viscosity" is probably not an accurate expression. Is the ileum flexible? Is it the viscosity of the gastrointestinal contents (ileal supernatant)?

Revised as ‘viscosity of the ileal supernatant’ (L 609-610).

L691-692 What does growth performance mean? What is the positive influence?

Revised as ‘increased body weight gain and feed conversion’ (L 696).

L687, L714

 I don't think "performance variables" and "production cycle" are appropriate words to use in this context. Specific items used as performance indicators should be shown individually.

Also, the term production cycle used here seems to refer to the rearing period, inferring from the context.

The terms ‘body weight gain, feed intake and feed efficiency’ was used instead of ‘performance variables’ (L 691, 719-720).

The term ‘production cycle’ was removed and the sentence was revised (L 693).

---

## [Decision Letter · Decision Letter 3]

21 Apr 2021

Effects of exogenous β-glucanase on ileal digesta soluble β-glucan molecular weight, digestive tract characteristics, and performance of coccidiosis vaccinated broiler chickens fed hulless barley-based diets with and without medication

PONE-D-20-20511R3

Dear Dr. Newkirk,

We’re pleased to inform you that your manuscript has been judged scientifically suitable for publication and will be formally accepted for publication once it meets all outstanding technical requirements.

Kind regards,

Arda Yildirim, Ph.D.

Academic Editor

PLOS ONE

Additional Editor Comments (optional):

Thank you for addressing all comments.

Reviewers' comments:

Reviewer's Responses to Questions

**Comments to the Author**

1. If the authors have adequately addressed your comments raised in a previous round of review and you feel that this manuscript is now acceptable for publication, you may indicate that here to bypass the “Comments to the Author” section, enter your conflict of interest statement in the “Confidential to Editor” section, and submit your "Accept" recommendation.

Reviewer #1: All comments have been addressed

2. Is the manuscript technically sound, and do the data support the conclusions?

Reviewer #1: Yes

3. Has the statistical analysis been performed appropriately and rigorously? 

Reviewer #1: Yes

4. Have the authors made all data underlying the findings in their manuscript fully available?

Reviewer #1: Yes

5. Is the manuscript presented in an intelligible fashion and written in standard English?

Reviewer #1: Yes

6. Review Comments to the Author

Reviewer #1: (No Response)

7. PLOS authors have the option to publish the peer review history of their article (what does this mean?). If published, this will include your full peer review and any attached files.

Reviewer #1: **Yes: **Takeshi Kawasaki

---

## [Editor Report · Acceptance letter]

23 Apr 2021

PONE-D-20-20511R3 

Effects of exogenous β-glucanase on ileal digesta soluble β-glucan molecular weight, digestive tract characteristics, and performance of coccidiosis vaccinated broiler chickens fed hulless barley-based diets with and without medication 

Dear Dr. Newkirk:

I'm pleased to inform you that your manuscript has been deemed suitable for publication in PLOS ONE. Congratulations! Your manuscript is now with our production department. 

Kind regards, 

on behalf of

Prof. Dr. Arda Yildirim 

Academic Editor

PLOS ONE